# PX4 Simulation Results of a Quadcopter with a Disturbance-Observer-Based and PSO-Optimized Sliding Mode Surface Controller

Yutao Jing [1,†], Xianghe Wang [2,†], Juan Heredia-Juesas [2], Charles Fortner [2], Christopher Giacomo [3], Rifat Sipahi [1] and Jose Martinez-Lorenzo [1,4,*]

1. Department of Mechanical & Industrial Engineering, Northeastern University, Boston, MA 02115, USA
2. Conducted Research While at Department of Mechanical & Industrial Engineering, Northeastern University, Boston, MA 02115, USA
3. Air Force Research Laboratory, Rome, NY 13441, USA
4. Department of Electrical & Computer Engineering, Northeastern University, Boston, MA 02115, USA
* Correspondence: j.martinez-lorenzo@northeastern.edu
† These authors contributed equally to this work.

**Abstract:** This work designed a disturbance-observer-based nonlinear sliding mode surface controller (SMC) and validated the controller using a simulated PX4-conducted quadcopter. To achieve this goal, this research (1) developed a dynamic mathematical model; (2) built a PX4-based simulated UAV following the model-based design process; (3) developed appropriate sliding mode control laws for each degree of freedom; (4) implemented disturbance observers on the proposed SMC controller to achieve finer disturbance rejection such as crosswind effect and other mutational disturbances; (5) optimized the SMC controller's parameters based on particle swarm optimization (PSO) method; and (6) evaluated and compared the quadcopter's tracking performance under a range of noise and disturbances. Comparisons of PID control strategies against the SMC were documented under the same conditions. Consequently, the SMC controller with disturbance observer facilitates accurate and fast UAV adaptation in uncertain dynamic environments.

**Keywords:** sliding-mode surface control; disturbance observer; nonlinear control; particle-swarm optimization; PX4; simulation; trajectory tracking; wind resistance

## 1. Introduction

An unmanned aerial vehicle (UAV) is an aircraft that does not have any human pilot, crew, or passengers on board. Quadrotor, a type of UAV, usually designed in an X configuration, with two diagonal motors rotating in one direction and two other diagonal motors rotating in the opposite direction to balance the torque. Quadrotors are under-actuated systems with four control inputs but six degrees of freedom. The attitude and altitude of the vehicle can be adjusted by controlling the four motors using pulse-width modulation (PWM) to increase and decrease thrust. Quadrotors are being used in an increasing number of civilian applications, including data collection, product delivery, and rescue searches [1,2]. In agriculture, quadcopters are used to spray pesticides and implement artificial rainfall [3]. In addition, UAVs have applications and developments in the fields of transportation, aerial photography, disaster relief, and artificial intelligence [4]. In the military field, quadcopters are used for combat, reconnaissance, patrol, and defense [5].

With the rapid development of science and technology, more and more disciplines and topics related to UAVs have been developed, such as UAV control [6] and UAV aerodynamics [7]. Recent research in the field of UAV control has progressed in several directions. Linear controllers have been proposed to control the position and the attitude of UAVs, for example, the Linear Quadratic Regulator (LQR) [6,8,9]. LQR is an optimal control strategy, primarily for linear systems, whereby a controller is designed by optimizing certain



performance indicators such as control effort and state variables. The control law of LQR has a linear relationship with the state of the system and is applicable also to multi-input and multi-output systems (MIMO). However, the LQR design may require an accurate system model, may lack robustness, and would be more feasible for linear models [10].

Although Proportional Integral Derivative (PID) control is more than 100 years old, it is still the most commonly used control method for UAVs due to its simplicity, intuitiveness, reliable performance, and ease of adjustment. However, PID controllers do not perform well in the face of high-frequency noise and external disturbances due to their simplified linear model and simple structure. Therefore, many researchers have conducted experiments and proposed modifications over the traditional PID controller. For example, Goel et al. [11] introduces an adaptive Digital PID Control for unknown dynamics problems, and Dong and He [12] propose a fuzzy PID method for quadrotor aircraft. Meanwhile, to improve the robustness of the controller in the face of wind disturbances, Zhou et al. [13] uses a cascade PID attitude control in the inner-loop control of the UAV. The automatic PID gain tuned by reinforcement learning neural networks is also mentioned within the field of UAV control [14].

In addition, some researchers have proposed the backstepping method. Backstepping is a recursive design method, applicable to both linear control and nonlinear control. The main idea of this method is to obtain a feedback controller by recursively constructing the Lyapunov function of the system and selecting an appropriate control law so that the Lyapunov function is bounded along the trajectory of the closed-loop system and the trajectory tends to be stable. The controller is designed without excessive simplification of the model, and the control of the UAV is more accurate, and the response speed is improved [15]. However, due to its weak disturbance-rejection capabilities, researchers usually combine the backstepping control with other methods such as disturbance observer-based control [16] and adaptive backstepping control [17].

The sliding mode control (SMC) has been proven to be a robust way to maintain the stability of an unknown UAV dynamics [18,19]. Much work has been conducted in this area as well, for example, in [20] where researchers verified the second-order sliding-mode approach and implemented it on a fixed-wing micro aerial vehicle (MAV). The study verifies that the performance of the sliding-mode controller exceeds a benchmark classical controller. The basic idea of sliding-mode control is to design the switching hyperplane (sliding mode plane) according to the dynamic characteristics of the system and then force the system state to converge from outside the hyperplane to the switching hyperplane by the sliding-mode controller. Sliding-mode control can overcome uncertainties of the system and provide provable stability in the control of nonlinear systems [21]. Moreover, it has the characteristics of having a fast response, simple operation, and robustness to external environmental disturbances [22].

Other research on UAVs, such as attitude estimation for collision recovery [23], bidirectional thrust for aggressive, and inverted quadrotor flight [24]; swift maneuvers [25]; and ROS-based trajectory generation [26] have also been developed. A controller-tuning strategy for the MAV carrying a cable-suspended load is proposed in [27], which finds a reasonable trade-off between the fast displacements of the MAV and well-damped oscillations of the load. On the other hand, the system may lose robustness to non-ideal dynamics and uncertainty.

In general, a controller that can be adopted into a real-world autopilot requires far more adaptability, robustness, and sophistication. This is because it should handle the non-linearities of the UAV, under-actuation limitations, as well as sensor dynamics, sensor noise, and disturbances in the surrounding environment [28]. In this study, we chose the PX4 autopilot as the UAV simulation platform. Some previous studies have validated the advantages and feasibility of PX4 in developing new flight controllers. For example, Gomez et al. developed a new PX4 Optimal PID Tuning method [29], Saengphet et al. focused on PX4 implemented PID control [30], and Niit and Smit validated the PX4 autopilot architecture with adaptive control and demonstrated the robustness of the controller under ideal conditions [31].

Nowadays, we still have a wide variety of quadrotor control problems, since nonlinearity and inter-dependency of UAV state variables within the mathematical model, as well as sensor implementations, noise, Kalman filters, and GPS delays, combined together are the main challenges in the design of fast responding, accurately tracking UAV controllers. In addition, the robustness of UAVs becomes a stringent requirement, in the face of turbulence effects and increasingly diverse external disturbances such as wind gusts [13] and additional loads [27]. Given the many strengths of sliding-mode controllers (SMC), this work focuses on the design of SMC that can be embedded into the PX4 framework. Simulation results show that SMC control outperforms PID control in many aspects, especially under large sensor noise and different types of random disturbances.

The design process of the SMC controller can be divided into six parts. (1) The development of a mathematical model in the MATLAB for the theory validation; (2) building a general PX4-based quadrotor as well as the test sites in a high-quality simulator; (3) developing a 6-degrees-of-freedom sliding-mode surface controller, which is a control scheme that includes an outer loop for position control and an inner loop for attitude control; (4) developing a 6-degreed-of-freedom disturbance observer that detects disturbance forces in the NED frame XYZ and disturbance moments around the NED frame XYZ axis; (5) optimizing SMC parameters based on an offline particle swarm optimization (PSO) method, a process that improves transient performance while reducing response time; and (6) verifying and comparing SMC against benchmark PID controllers in terms of settling time, overshoot, rise time, and tracking performance.

The article is presented in five sections. In Section 2, some assumptions and an introduction about the PX4 autopilot and baseline PID controllers (e.g., PID Position only controller, PID Rate based controller) will be described. The dynamic equations of quadrotors will be listed. The details of the SMC control structure and the disturbance observer will also be derived. Finally, the PSO algorithm will be implemented to optimize the SMC parameters. In Section 3, time simulations of UAV maneuvers are evaluated with SMC and PID controllers, including controllers optimized with PSO, and performances of these controllers are documented under noisy sensors and wind disturbances. In Section 4, we discuss the results, and, in Section 5, some future directions will be presented.

## 2. Materials and Methods

Computer simulation is used in many research areas to reduce experimental costs as well as the research and development cycle [10]. PX4 is an autopilot flight-control architecture and jMAVsim is the simulator we used in this study. One advantage of using PX4 and jMAVsim is that they provide a comprehensive model of the UAV and its sensors, as well as the UAV environment, such as wind speeds.

Table 1 shows the sensor properties: GPS interval (GPS_I), GPS Delay (GPS_D), GPS Start time (GPS_S), Accelerometer Noise (Acc_N), Gyroscope Noise (Gyr_N), Magnetometer Noise (Mag_N), and Pressure Sensor Noise (Prs_N), and Table 2 introduces the initial Magnetometer condition, including the set values and their deviations (N: north; E: east; D: down; Incl: Magnetic inclination; Decl: Magnetic declination; H-Magn: Magnetic field strength; T-Magn: Magnetic flux density). Table 3 displays the wind conditions (N: north; E: east; D: down; SET: current wind; DEV: deviation), and, in the beginning of simulations, the airflow speed has been set to zero, which will be adjusted in the wind-resistance-testing section. The final Table 4 gives the home position as well as the gravity setting (LAT: latitude; LON: longitude; ALT: altitude; Grav: Gravity; INIT: initial home position).

**Table 1.** Sensors (GPS, Accelerometer, Gyroscope, Magnetometer, Pressure Sensor) configuration.

| Sensors Properties | $GPS_I$ | $GPS_D$ | $GPS_S$ | $ACC_N$ | $GYO_N$ | $MAG_N$ | $PRS_N$ |
|---|---|---|---|---|---|---|---|
| Value | 50 | 300 | 10 | 0.05 | 0.01 | 0.005 | 0.1 |
| Unit | ms | ms | ms | $m/s^2$ | rad/s | Gauss | m |

**Table 2.** Magnetic field settings.

| Mag-Field Settings | *N* | *E* | *D* | Incl | Decl | H-Magn | T-Magn |
|---|---|---|---|---|---|---|---|
| Value | 0.4487 | 0.01669 | 0.89353 | 63.33588 | 2.13 | 0.44901 | 1 |
| Unit | Gauss | Gauss | Gauss | deg | deg | oersteds | Gauss |

**Table 3.** Initial wind settings.

| Wind Settings | N-SET | E-SET | D-SET | N-DEV | E-DEV | D-DEV |
|---|---|---|---|---|---|---|
| Value | 0 | 0 | 0 | 0 | 0 | 0 |
| Unit | m/s | m/s | m/s | m/s | m/s | m/s |

**Table 4.** Initial position settings.

| Home Position | LAT | LON | ALT | INIT-N | INIT-E | INIT-D | Grav |
|---|---|---|---|---|---|---|---|
| Value | 55.753395 | 37.625427 | 155 | 0 | 0 | 0 | 9.81 |
| Unit | deg | deg | m | m | m | m | m/s$^2$ |

The general approach in this manuscript can be summarized as follows: We first utilized the mathematical model of the UAV to design the inner/outer loop SMC. The UAV model as well as the SMC architecture is next implemented in MATLAB/SIMULINK to verify the reliability of SMC. This architecture also includes disturbance observers who are able to compensate against disturbances. Benchmark PID controllers are also implemented in these simulations. Next, PSO was used as an offline optimization tool to intelligently iterate the controller's parameters to improve, for example, the settling time, overshoot, and tracking performance.

Once the SMC and PID controllers performed as expected in the ideal mathematical model, we next port them to the PX4-based simulated UAV as shown in Figure 1, where we can simulate and evaluate the UAV flight dynamics, by customizing the physical parameters of the UAV, the noise range of the sensors, the ambient magnetic field, and the environmental disturbances. Finally, we can quantify the performance of SMC and PID based on simulation data collected from PX4 flight tests.

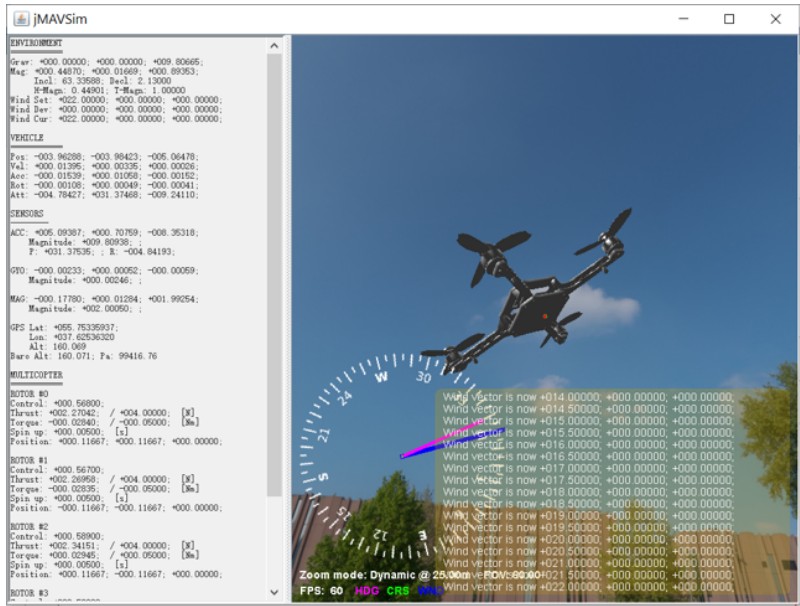

**Figure 1.** Simulated PX4-based quadcopter flying in the windy jMavsim simulator environment.

### 2.1. Baseline PID Controllers and Quadrotor Dynamics Equations

Two predefined proportional-integral-derivative (PID) controllers were chosen as the baseline and compared with SMC. The first PID controller is a pure-position PID controller, which is widely used for the position control of quadcopters [32]. This controller can track the desired attitude, including the pitch, roll, and yaw, based on only position errors. In its control loop, shown in Figure 2, three separate PID modules compare the difference between the current XYZ position values and the desired XYZ position values, and those PID for XY positions help generate the desired rotation angles for roll and pitch. In our simulations, we limit the maximum rotation angle in roll and pitch to 50 degrees. The inputs to the PID controllers are errors in $X$, $Y$, $Z$, $Yaw$, $Pitch$, and $Roll$. The outputs are virtual control signals $\tau_{Thrust}$, $\tau_{Pitch}$, $\tau_{Roll}$, $\tau_{Yaw}$, which can be converted to PWM signals in four channels to the motors via the mixer.

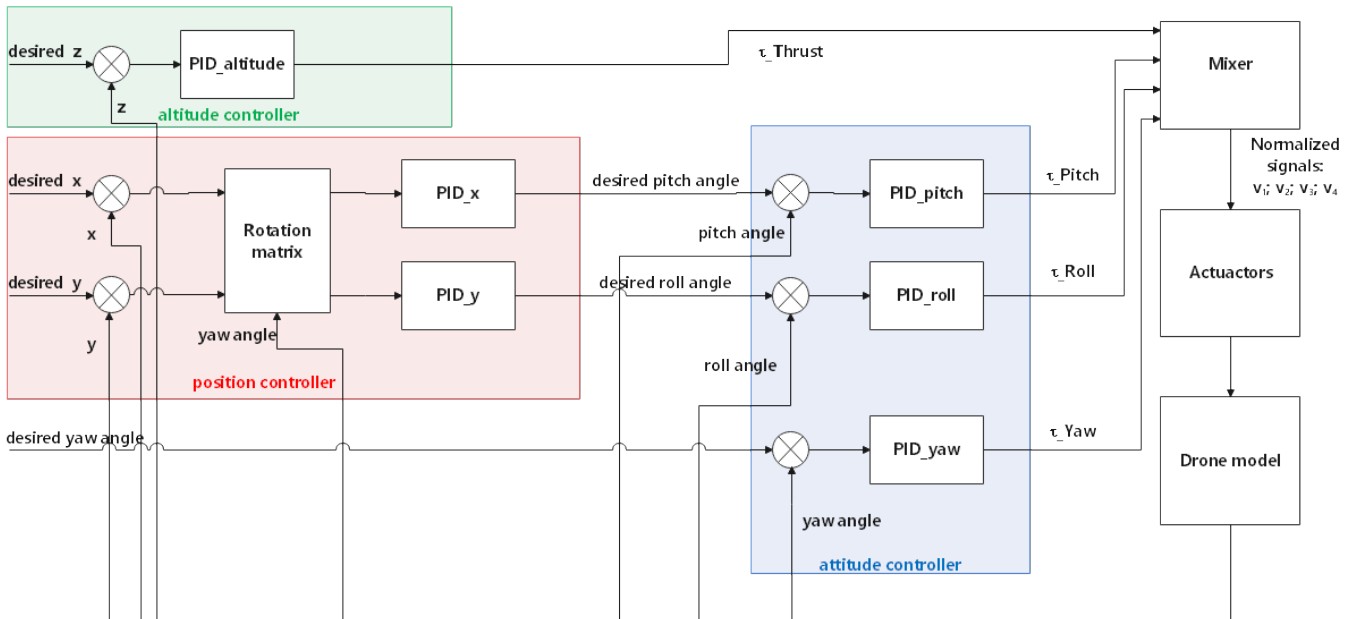

**Figure 2.** The PID-position-only controller diagram.

The second PID controller is a rate controller shown in Figure 3, which controls the speed of the UAV in addition to position tracking, so this PID controller has the potential to offer greater flight dynamic responses, as it acts on the "rate" [33]. In short, the rate controller tracks the desired trajectory based on the velocity error, while the pure position controller tracks based on position error. In our experiments, the velocity limits are set to 20 m/s in the XY direction and 10 m/s in the Z direction. The inputs to the controllers are errors in $X$, $V_x$, $Y$, $V_y$, $Z$, $V_z$, $Yaw$, $Pitch$, and $Roll$. The outputs are virtual control signals, which are converted to PWM signals in four channels to the motors via the mixer.

We use the X mode quadcopter drone in PX4, as shown in Figure 4, and NED (north, east, and down) as the global coordinate system. The UAV dynamics equations are expressed as [34]:

$$\begin{cases} \ddot{x} = -\frac{1}{m}[(\sin\phi\sin\psi + \cos\phi\sin\theta\cos\psi) \cdot (T_1 + T_2 + T_3 + T_4) + K_1 \cdot \dot{x} \cdot |\dot{x}|] \\ \ddot{y} = -\frac{1}{m}[(-\sin\phi\cos\psi + \cos\phi\sin\theta\sin\psi) \cdot (T_1 + T_2 + T_3 + T_4) + K_1 \cdot \dot{y} \cdot |\dot{y}|] \\ \ddot{z} = -\frac{1}{m}[(\cos\phi\cos\theta)(T_1 + T_2 + T_3 + T_4) + K_1 \cdot \dot{z} \cdot |\dot{z}|] + g \\ \dot{p} = \frac{1}{J_x}[\frac{\sqrt{2}}{2}l(T_2 + T_3 - T_1 - T_4) - q \cdot r(J_z - J_y) - K_2 \cdot p \cdot |p|] \\ \dot{q} = \frac{1}{J_y}[\frac{\sqrt{2}}{2}l(T_1 + T_3 - T_2 - T_4) - p \cdot r(J_x - J_z) - K_2 \cdot q \cdot |q|] \\ \dot{r} = \frac{1}{J_z}[(Q_1 + Q_2 - Q_3 - Q_4) - p \cdot q(J_y - J_x) - K_2 \cdot r \cdot |r|] \\ \dot{\phi} = p + q \cdot \tan\theta\sin\phi + r \cdot \tan\theta\cos\phi \\ \dot{\theta} = q \cdot \cos\phi - r \cdot \sin\phi \\ \dot{\psi} = q\frac{\sin\phi}{\cos\theta} + r\frac{\cos\phi}{\cos\theta} \end{cases} \quad (1)$$

where the $x, y, z$ directions are in the global coordinate system, $x', y', z'$ are in drone's body coordinate system; $p, q, r$ are angular velocity in the drone body frame; $\phi, \theta, \psi$ are Euler angles; $m$ is the mass of the drone; $l$ is the distance between the center of the propeller and the center of the drone; $J_x, J_y, J_z$ are the moment of inertia; $T_{1-4}$ are the thrusts of each propeller; $Q_{1-4}$ are anti-torques of each motor given by the propellers; $K_1$ is the coefficient of air drag force; $K_2$ is the air drag torque coefficient; $T\_max$ is the full thrust of motor; and $Q\_max$ is anti-torque at the full thrust of the motor. $x, y, z, p, q, r, \phi, \theta, \psi$ are also shown in Figure 4. The aerodynamic forces are modeled as a nonlinear drag opposing the motion, and the aerodynamics moments are modeled as a nonlinear drag moment opposing the rotation:

$$\begin{cases} F_{drag} = -K_1 \cdot v \cdot |v| \\ M_{drag} = -K_2 \cdot w \cdot |w| \end{cases} \quad (2)$$

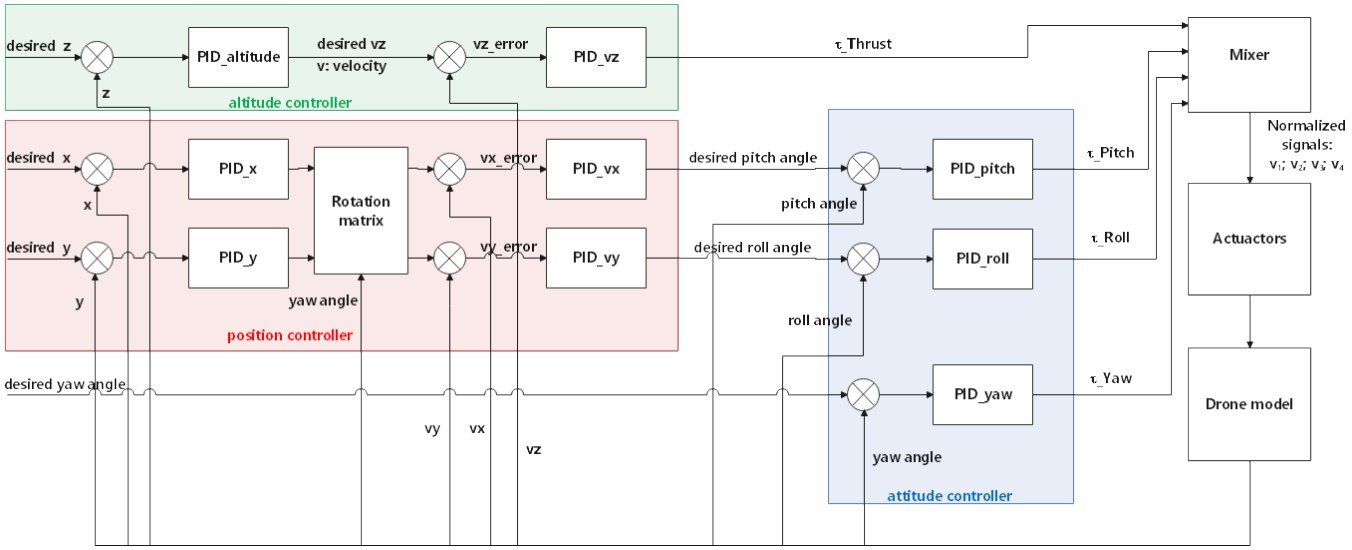

**Figure 3.** The PID-rate controller diagram.

The thrust of each propeller $T_{1-4}$ and torque $Q_{1-4}$ are determined by normalized signal $v_{1-4}$.

$$\begin{cases} T_{1-4} = T_{max} \cdot v_{1-4} \\ Q_{1-4} = Q_{max} \cdot v_{1-4} \end{cases} \quad (3)$$

where:

$$\begin{bmatrix} v_1 \\ v_2 \\ v_3 \\ v_4 \end{bmatrix} = (1000 \cdot \begin{bmatrix} 1 & -1 & 1 & 1 \\ 1 & 1 & -1 & 1 \\ 1 & -1 & -1 & -1 \\ 1 & 1 & 1 & -1 \end{bmatrix} \cdot \begin{bmatrix} \tau_T \\ \tau_P \\ \tau_R \\ \tau_Y \end{bmatrix} - P_{min}) \cdot \frac{1}{P_{max} - P_{min}}, \quad (4)$$

where $\tau_T, \tau_P, \tau_R,$ and $\tau_Y$ represent the $\tau_{Thrust}, \tau_{Pitch}, \tau_{Roll},$ and $\tau_{Yaw}$ virtual control inputs to control the thrust, pitch motion, roll motion, and yaw motion of the drone, respectively, the ranges of the virtual control inputs are set as: $1 \leq \tau_T \leq 2; -0.05 \leq \tau_P \leq 0.05; -0.05 \leq \tau_R \leq 0.05; -0.1 \leq \tau_Y \leq 0.1;$ and $P_{max}, P_{min}$, which are represented as $PWM\_max, PWM\_min$, which are the maximum and minimum values in the process of PWM normalization, respectively. Equation (4) is a normalization process, which is called "mixing" in PX4 architecture, the structure of the $4 \times 4$ matrix which provides the virtual control signals is decided by the motor number and propeller spin direction. Mixing means to receive commands (e.g., turn right) and translate them to actuator commands, here with regards to motors or servos. According to the PX4 definition, $PWM\_max = 2000$ and $PWM\_min = 1000$. Substituting these into Equation (4), we have:

$$\begin{bmatrix} v_1 + v_2 + v_3 + v_4 \\ v_2 + v_3 - v_1 - v_4 \\ v_1 + v_3 - v_2 - v_4 \\ v_1 + v_2 - v_3 - v_4 \end{bmatrix} = \begin{bmatrix} 4\tau_T - 4 \\ -4\tau_R \\ -4\tau_P \\ 4\tau_Y \end{bmatrix} \tag{5}$$

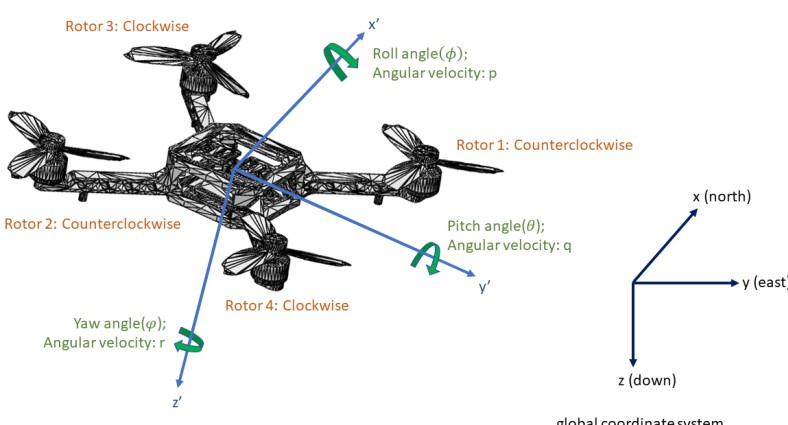

**Figure 4.** Euler angles and the coordinate system.

The physical parameters of the UAV dynamics model are shown in Table 5.

**Table 5.** UAV dynamics model parameters.

| Parameter | $m$ | $l$ | $J_x$ | $J_y$ | $J_z$ | $K_1$ | $K_2$ |
|---|---|---|---|---|---|---|---|
| Value | 0.8 | 0.165 | 0.005 | 0.005 | 0.009 | 0.01 | 0 |
| Unit | kg | $m$ | $kg \cdot m^2$ | $kg \cdot m^2$ | $kg \cdot m^2$ | N/(m/s) | N · m/(rad/s) |

| | $T_{max}$ | $Q_{max}$ | $P_{max}$ | $P_{min}$ | | | |
|---|---|---|---|---|---|---|---|
| | 4 | 0.05 | 2000 | 1000 | | | |
| | N | N · m | µs | µs | | | |

## 2.2. Sliding Mode Controller

Sliding-mode control (SMC), a type of variable structure control (VSC), is a nonlinear control method [22]. The non-linearity is created by the discontinuity in the control law as explained next. The sliding-mode control designs a sliding surface (usually expressed as $s = 0$), and it has different control actions, usually expressed as $u = \begin{cases} u+ \\ u- \end{cases}$ , to make the system states approach to the sliding surface depending on the sign of $s$. SMC has the advantages of a fast response, robustness, and handling nonlinear dynamical systems. However, when the state trajectory reaches the sliding surface, instead of sliding along the surface perfectly, it will rapidly switch between both sides of the surface. Hence, it is common for SMC to have chattering problems.

The overall control structure of the SMC controller is shown in Figure 5. Each SMC is equipped with a disturbance observer.

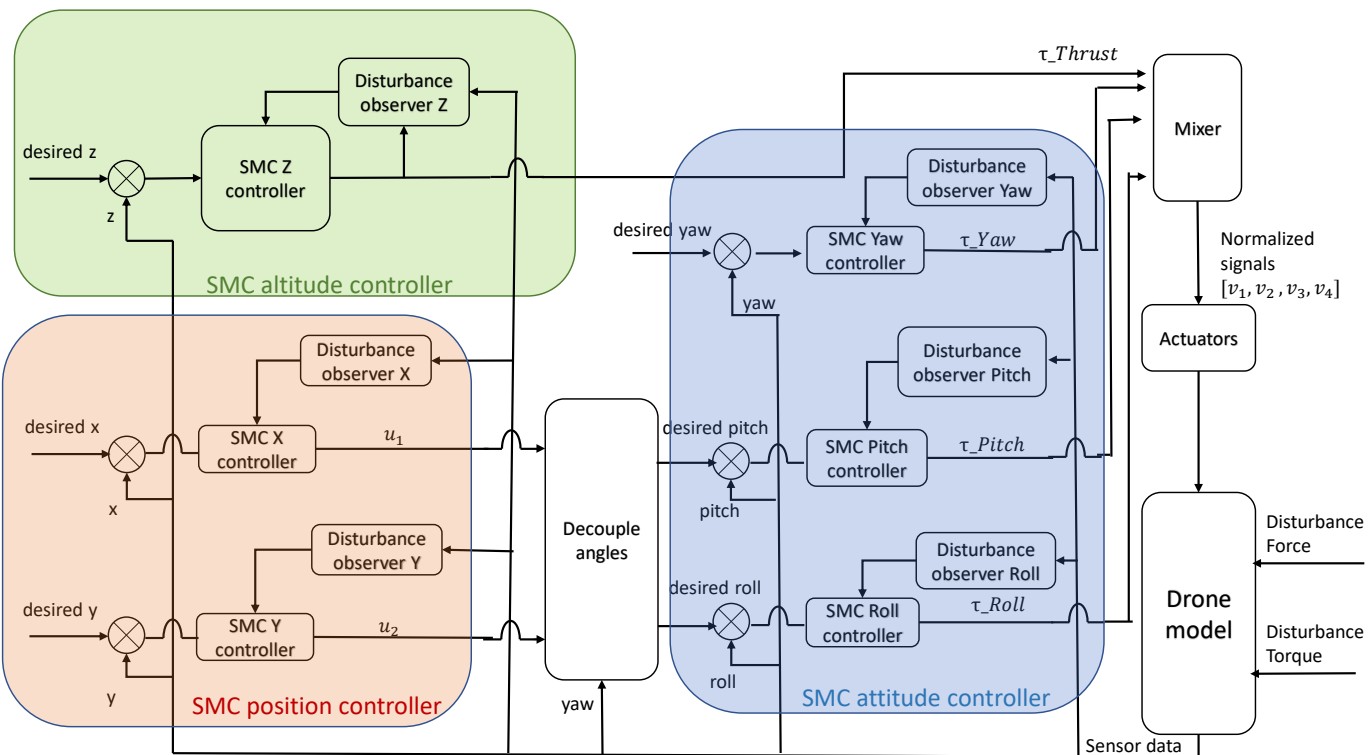

**Figure 5.** The SMC controller diagram.

### 2.2.1. SMC X-Y Position Controller

The position controllers of x and y are similar. Figure 6 shows an example of how the system approaches the desired sliding surface. Here the x and y axes are the error $e$ and derivative of error $\dot{e}$, respectively, and $X_0$ is the initial state, which means there is some error between the initial state and desired state. Naturally, the desired final value should be $(e, \dot{e}) = (0,0)$, which means the error and the derivative of the error must both go to zero, and the system reaches its steady state. Moreover, the blue line is the sliding surface that should be designed, and the system states should approach this line and slide over the line toward $(e, \dot{e}) = (0,0)$.

In this process, the motion of the system state can be divided into two types: one is the reaching mode (the process that the system state leaves $X_0$ and moves toward the sliding surface), which is controlled by control law u; and the other is called sliding mode (the process that the system state approaches the desired value along the sliding surface and remains on this surface, moving toward the origin $(0,0)$), which is defined by the sliding-mode surface. To obtain high performance of the state trajectories, we should carefully design the characters of both the control laws and the sliding surfaces. Accordingly, the design of the sliding mode controller is performed in two steps. A reasonable sliding surface makes the system to have quality performance in the sliding mode, and the design of the control law $u$ needs to ensure that the system can reach the sliding-mode surface from any state. According to (1):

$$\ddot{x} = -\frac{1}{m}[u_1 \cdot \boldsymbol{Thrust} + K_1 \cdot \dot{x} \cdot |\dot{x}|], \tag{6}$$

where $u_1 = \sin\phi\sin\psi + \cos\phi\sin\theta\cos\psi$ is the control signal and $\boldsymbol{Thrust} = T_1 + T_2 + T_3 + T_4$. The sliding surface is selected as:

$$s = a_1 e + \dot{e}, \tag{7}$$

where $a_1 > 0$, and $e = x - des\_x$ is the tracking error. To perform the stability analysis, we choose $V(s) = \frac{1}{2}s^2 > 0$ as the Lyapunov function. If $\dot{V}(s) < 0$, then the system is stable, since this implies $s$ approaches 0 (i.e., $e \to 0$ and $\dot{e} \to 0$). To this end, we obtain $\dot{V}(s)$ as:

$$
\begin{aligned}
\dot{V}(s) &= s\dot{s} = s(a_1\dot{e} + \ddot{e}) = s(a_1\dot{e} + \ddot{x} - d\ddot{e}s\_x) \\
&= s[a_1\dot{e} - \frac{1}{m}(u_1 \cdot \textbf{\textit{Thrust}} + K_1 \cdot \dot{x} \cdot |\dot{x}|) - d\ddot{e}s\_x].
\end{aligned}
\tag{8}
$$

Next, we propose the control law:

$$
u_1 = k_1' \cdot \frac{m}{\textbf{\textit{Thrust}}} \cdot sat(\frac{s}{\epsilon}) + k_2 \cdot \frac{m}{\textbf{\textit{Thrust}}} \cdot \dot{e} + c \cdot \frac{m}{\textbf{\textit{Thrust}}} \cdot s - \frac{K_1 \cdot \dot{x} \cdot |\dot{x}|}{\textbf{\textit{Thrust}}} - \frac{m \cdot d\ddot{e}s\_x}{\textbf{\textit{Thrust}}},
\tag{9}
$$

where:

$$
k_1' = \begin{cases} k_1, & (b_1 \cdot |e| + b_2 \cdot |\dot{e}|) < k_1 \\ (b_1 \cdot |e| + b_2 \cdot |\dot{e}|), & (b_1 \cdot |e| + b_2 \cdot |\dot{e}|) \geq k_1 \\ . \end{cases}
\tag{10}
$$

In (9), the saturation function is used to solve the chattering problem due to the switches in the sign of $s$:

$$
sat(\frac{s}{\epsilon}) = \begin{cases} \frac{s}{\epsilon}, & |s| < \epsilon \\ sign(\frac{s}{\epsilon}) = \begin{cases} 1, & s \geq \epsilon \\ -1, & s \leq -\epsilon \end{cases}, & |s| \geq \epsilon, \end{cases}
\tag{11}
$$

where $\epsilon > 0$ is a user-defined constant value. Here, we use $\epsilon = 0.5$. Inserting (9) into (8), we obtain:

$$
\dot{V}(s) = s[(a_1 - k_2) \cdot \dot{e} - k_1' \cdot sat(\frac{s}{\epsilon}) - c \cdot s].
\tag{12}
$$

From Equation (12), we can derive that $\begin{cases} a_1 - k_2 = 0 \\ k_1' > 0 \\ c > 0 \end{cases}$ is the sufficient condition for $\dot{V}(s) < 0$ to hold. These are the conditions for stability in the x direction. When optimizing the SMC using PSO in Section 2.4, these conditions will be incorporated into PSO to make sure PSO will optimize the drone dynamics while respecting theoretical stability.

Figure 6 shows how $e$ and $\dot{e}$ reach the sliding surface. The data used in this figure are from one of the simulation tests conducted with PX4. Due to the aforementioned sensor noise, we observe from the zoom-in figure that $e$ and $\dot{e}$ do not reach (0,0) perfectly.

Due to the outputs of the SMC, the position controller should be desired values of roll angle and pitch angle to be passed to the attitude controller, we need to calculate the desired $\phi$ and $\theta$, see Figure 5. Recall that we have $u_1 = (\sin\phi \sin\psi + \cos\phi \sin\theta \cos\psi)$ from the x controller, and, similarly, we use the same SMC controller structure for the y direction, where it is easy to show that the control law will be $u_2 = (-\sin\phi \cos\psi + \cos\phi \sin\theta \sin\psi)$. Then, using $u_1$, $u_2$, we derive the desired $\phi$ and $\theta$ quantities as:

$$
\begin{cases} \phi = \arcsin(u_1 \sin\psi - u_2 \cos\psi) \\ \theta = \arcsin\frac{u_1 \cos\psi + u_2 \sin\psi}{\cos\phi}. \end{cases}
\tag{13}
$$

The above quantities are denoted below by $des\_\phi$ and $des\_\theta$, respectively.

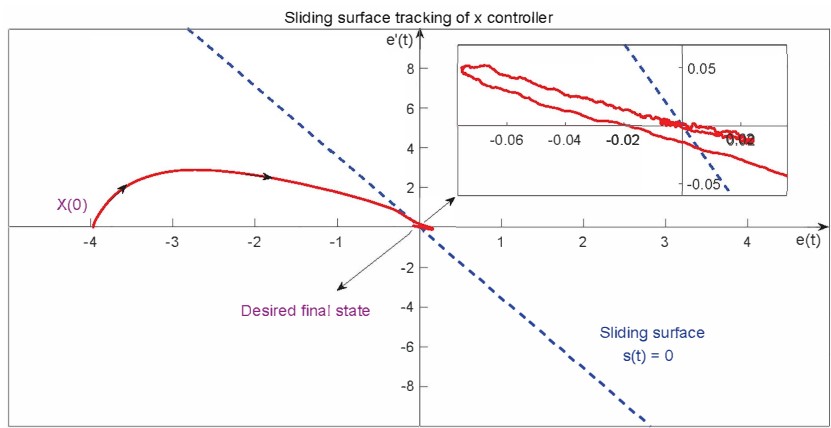

**Figure 6.** Sliding-surface tracking of x position controller on the plane of $e$ vs. $\dot{e}$.

### 2.2.2. SMC Altitude Controller

According to Equations (1), (3), and (5):

$$\ddot{z} = -\frac{1}{m}[(\cos\phi\cos\theta) \cdot T_{max}(4\tau_T - 4) + K_1 \cdot \dot{z} \cdot |\dot{z}|] + g, \tag{14}$$

and $\tau_T$ is the control signal. The sliding surface is designed as:

$$s = a_2 e + \dot{e}, \tag{15}$$

where $a_2 > 0$, and $e = z - des\_z$. Let the control law be:

$$\tau_T = \frac{m}{4\cos\theta\cos\phi \cdot T_{max}} \cdot k_3' \cdot sat(\frac{s}{\epsilon}) + \frac{k_4 \cdot \dot{e}}{\cos\theta\cos\phi} + \frac{-K_1 \cdot \dot{z} \cdot |\dot{z}| + mg - m \cdot de\ddot{s}\_z}{4T_{max} \cdot \cos\phi\cos\theta} + 1, \tag{16}$$

where:

$$k_3' = \begin{cases} k_3, & (b_3 \cdot |e| + b_4 \cdot |\dot{e}|) < k_3 \\ (b_3 \cdot |e| + b_4 \cdot |\dot{e}|), & (b_3 \cdot |e| + b_4 \cdot |\dot{e}|) \geq k_3 \\ , \end{cases} \tag{17}$$

with $k_3 > 0$ and $k_4 > 0$. Proposing again $V(s) = \frac{1}{2}s^2 > 0$, it is possible to render $\dot{V}(s) < 0$ for $\tau_T$ in (16). To show this, we calculate $\dot{V}(s)$ as:

$$\begin{aligned} \dot{V}(s) &= s\dot{s} = s(a_2\dot{e} + \ddot{z} - de\ddot{s}\_z) \\ &= s[(a_2 - \frac{4T_{max}}{m} \cdot k_4)\dot{e} - \frac{4\cos\phi\cos\theta \cdot T_{max}}{m} \cdot k_3' \cdot sat(\frac{s}{\epsilon})], \end{aligned} \tag{18}$$

Hence, $\begin{cases} a_2 - \frac{4T_{max}}{m} \cdot k_4 = 0 \\ k_3' > 0 \end{cases}$ satisfy $\dot{V}(s) < 0$.

In Figure 7, we show how $e$ and $\dot{e}$ approach the sliding surface in the z direction in one of the simulation tests.

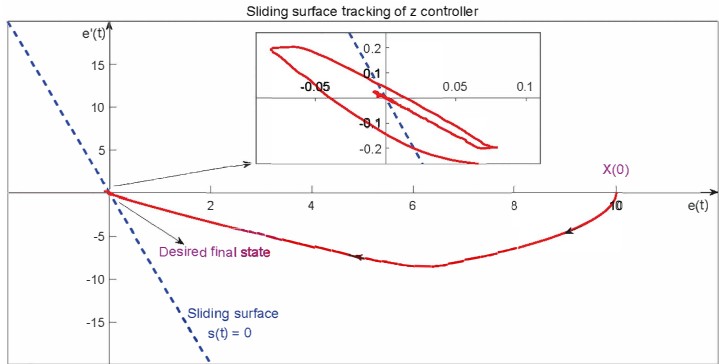

**Figure 7.** Sliding-surface tracking of z altitude controller on the plane of $e$ vs. $\dot{e}$.

### 2.2.3. SMC Attitude Controller

Different from the position control, as for the attitude controllers, the reaching law is designed as the following isokinetic form: $\dot{s} = -c \cdot sgn(s)$. Firstly, we present the roll-angle control. According to Equations (1), (3), and (5), we derive the following relationship:

$$
\begin{aligned}
\ddot{\phi} =& \dot{p} + \dot{q} \cdot \tan\theta \sin\phi + q \cdot (\frac{1}{\cos^2\theta} \cdot \dot{\theta} \cdot \sin\phi + \tan\theta\cos\phi \cdot \dot{\phi}) + \\
& \dot{r} \cdot \tan\theta\cos\phi + r \cdot (\frac{1}{\cos^2\theta} \cdot \dot{\theta} \cdot \cos\phi - \tan\theta\sin\phi \cdot \dot{\phi}) \\
=& -\frac{2\sqrt{2}\cdot l \cdot T_{max}}{J_x} \cdot \tau_R - \frac{2\sqrt{2}\cdot l \cdot T_{max} \cdot \tan\theta\sin\phi}{J_y} \cdot \tau_P + \frac{4Q_{max}\tan\theta\cos\phi}{J_z} \cdot \tau_Y + f_\phi,
\end{aligned}
$$ (19)

where $f_\phi$:

$$
\begin{aligned}
f_\phi =& q \cdot (\frac{1}{\cos^2\theta} \cdot \dot{\theta} \cdot \sin\phi + \tan\theta\cos\phi \cdot \dot{\phi}) + r \cdot (\frac{1}{\cos^2\theta} \cdot \dot{\theta} \cdot \cos\phi - \tan\theta\sin\phi \cdot \dot{\phi}) \\
& - \frac{J_z - J_y}{J_x} \cdot qr - \frac{K_2}{J_x} \cdot p \cdot |p| - \frac{J_x - J_z}{J_y} \cdot pr \cdot \tan\theta\sin\phi - \frac{K_2}{J_y} \cdot q \cdot |q| \cdot \tan\theta\sin\phi \\
& - \frac{J_y - J_x}{J_z} \cdot pq \cdot \tan\theta\cos\phi - \frac{K_2}{J_z} \cdot r \cdot |r| \cdot \tan\theta\cos\phi.
\end{aligned}
$$ (20)

Next, we propose a sliding surface:

$$
s_\phi = a_3 e_\phi + \dot{e}_\phi,
$$ (21)

where $a_3 > 0$ and $e_\phi = \phi - des\_\phi$. The control law:

$$
\begin{aligned}
\tau_R =& k_5 \cdot sat(\frac{s_\phi}{\epsilon}) + k_6 \cdot \dot{e}_\phi - \frac{J_x}{J_y} \cdot \tan\theta\sin\phi \cdot \tau_P + \frac{4J_x \cdot Q_{max} \cdot \tan\theta\cos\phi}{2\sqrt{2}J_z \cdot l \cdot T_{max}} \cdot \tau_Y \\
& + \frac{J_x}{2\sqrt{2}\cdot l \cdot T_{max}} \cdot (f_\phi - de\ddot{s}\_\phi)
\end{aligned}
$$ (22)

can guarantee stability. To show this, let $V(s) = \frac{1}{2}s^2 > 0$. Then:

$$
\begin{aligned}
\dot{V}(s) =& s_\phi \dot{s}_\phi = s_\phi(a_3\dot{e}_\phi + \ddot{\phi} - de\ddot{s}\_\phi) \\
=& s_\phi[(a_3 - \frac{2\sqrt{2}\cdot l \cdot T_{max}}{J_x} \cdot k_6)\dot{e}_\phi - \frac{2\sqrt{2}\cdot l \cdot T_{max}}{J_x} \cdot k_5 \cdot sat(\frac{s_\phi}{\epsilon})].
\end{aligned}
$$ (23)

Thus, we obtain $\begin{cases} a_3 - \frac{2\sqrt{2}T_{max}l}{J_x}k_6 = 0 \\ k_5 > 0 \end{cases}$ to make $\dot{V}(s) < 0$.

Secondly, for the pitch-angle controller, consider similarly,

$$\ddot{\theta} = \dot{q} \cdot \cos\phi - q \cdot \sin\phi \cdot \dot{\phi} - \dot{r} \cdot \sin\phi - r \cdot \cos\phi \cdot \dot{\phi}$$
$$= -\frac{\cos\phi \cdot 2\sqrt{2} \cdot l \cdot T_{max}}{J_y} \cdot \tau_P - \frac{4Q_{max}\sin\phi}{J_z} \cdot \tau_Y + f_\theta, \tag{24}$$

where $f_\theta$:

$$f_\theta = -\left(q \cdot \sin\phi \cdot \dot{\phi} + r \cdot \cos\phi \cdot \dot{\phi} + \frac{J_x - J_z}{J_y} \cdot pr \cdot \cos\phi + \frac{K_2}{J_y} \cdot q \cdot |q| \cdot \cos\phi\right)$$
$$+ \frac{J_y - J_x}{J_z} \cdot pq \cdot \sin\phi + \frac{K_2}{J_z} \cdot r \cdot |r| \cdot \sin\phi. \tag{25}$$

We next propose the sliding surface:

$$s_\theta = a_3 e_\theta + \dot{e}_\theta, \tag{26}$$

with $a_3 > 0$ and $e_\theta = \theta - des\_\theta$. The control law:

$$\tau_P = \frac{1}{\cos\phi} \cdot k_5 \cdot sat\left(\frac{s_\theta}{\epsilon}\right) + \frac{k_6}{\cos\phi} \cdot \dot{e}_\theta - \frac{4J_y \cdot Q_{max} \cdot \tan\phi}{2\sqrt{2}J_z \cdot l \cdot T_{max}} \cdot \tau_Y$$
$$+ \frac{J_y}{\cos\phi \cdot 2\sqrt{2} \cdot l \cdot T_{max}} \cdot (f_\theta - des\ddot{\_}\theta) \tag{27}$$

can guarantee stability. To show this, let $V(s) = \frac{1}{2}s^2 > 0$, then:

$$\dot{V}(s) = s_\theta \dot{s}_\theta = s_\theta(a_3 \dot{e}_\theta + \ddot{\theta} - des\ddot{\_}\theta)$$
$$= s_\theta\left[\left(a_3 - \frac{2\sqrt{2} \cdot l \cdot T_{max}}{J_y} \cdot k_6\right)\dot{e}_\theta - \frac{\cos\theta \cdot 2\sqrt{2} \cdot l \cdot T_{max}}{J_y} \cdot k_5 \cdot sat\left(\frac{s_\theta}{\epsilon}\right)\right]. \tag{28}$$

Hence, we obtain $\begin{cases} a_3 - \frac{2\sqrt{2}T_{max}l}{J_y}k_6 = 0 \\ k_5 > 0 \end{cases}$　to make $\dot{V}(s) < 0$.

Thirdly, for the yaw angle, we have:

$$\ddot{\psi} = \dot{q} \cdot \frac{\sin\phi}{\cos\theta} + q \cdot \frac{\cos\phi\cos\theta \cdot \dot{\phi} + \sin\phi\sin\theta \cdot \dot{\theta}}{\cos^2\theta} +$$
$$\dot{r} \cdot \frac{\cos\phi}{\cos\theta} + r \cdot \frac{-\sin\phi\cos\theta \cdot \dot{\phi} + \cos\phi\sin\theta \cdot \dot{\theta}}{\cos^2\theta}$$
$$= \frac{\cos\phi \cdot 4Q_{max}}{\cos\theta \cdot J_z} \cdot \tau_Y - \frac{\sin\phi \cdot 2\sqrt{2} \cdot l \cdot T_{max}}{\cos\theta \cdot J_y} \cdot \tau_P + f_\psi, \tag{29}$$

where $f_\psi$:

$$f_\psi = q \cdot \frac{\cos\phi\cos\theta \cdot \dot{\phi} + \sin\phi\sin\theta \cdot \dot{\theta}}{\cos^2\theta} + r \cdot \frac{-\sin\phi\cos\theta \cdot \dot{\phi} + \cos\phi\sin\theta \cdot \dot{\theta}}{\cos^2\theta}$$
$$- \frac{\cos\phi(J_y - J_x)}{\cos\theta \cdot J_z} \cdot pq - \frac{\cos\phi \cdot K_2}{\cos\theta \cdot J_z} \cdot r \cdot |r| - \frac{\sin\phi(J_x - J_z)}{\cos\theta \cdot J_y} \cdot pr - \frac{\sin\phi \cdot K_2}{\cos\theta \cdot J_y} \cdot .q \cdot |q| \tag{30}$$

With the sliding surface:

$$s_\psi = a_4 e_\psi + \dot{e}_\psi, \tag{31}$$

where $a_4 > 0$ and $e_\psi = \psi - des\_\psi$. The control law:

$$\tau_Y = -\frac{\cos\theta}{\cos\phi} \cdot k_7 \cdot sat(\frac{s_\psi}{\epsilon}) - k_8 \cdot \frac{\cos\theta}{\cos\phi} \cdot \dot{e}_\psi + \frac{2\sqrt{2}J_z \cdot l \cdot T_{max} \cdot \tan\phi}{4J_y \cdot Q_{max}} \cdot \tau_P$$
$$- \frac{J_z \cdot \cos\theta}{4 \cdot Q_{max} \cdot \cos\phi} \cdot (f_\psi - de\ddot{s}\_\psi) \tag{32}$$

can guarantee stability. Stability analysis follows similarly, where we set $V(s) = \frac{1}{2}s^2 > 0$ and obtain:

$$\dot{V}(s) = s_\psi \dot{s}_\psi = s_\psi(a_4 \dot{e}_\psi + \ddot{\psi} - de\ddot{s}\_\psi)$$
$$= s_\psi[(a_4 - \frac{4Q_{max}}{J_z} \cdot k_8)\dot{e}_\psi - \frac{\cos\phi \cdot 4Q_{max}}{\cos\theta \cdot J_z} \cdot k_7 \cdot sat(\frac{s_\psi}{\epsilon})]. \tag{33}$$

Hence, we obtain $\begin{cases} a_4 - \frac{4Q_{max}}{J_z} \cdot k_8 = 0 \\ k_7 > 0 \end{cases}$ to make $\dot{V}(s) < 0$.

Finally, based on Equations (22), (27), and (32), we have:

$$\begin{bmatrix} \tau_R \\ \tau_P \\ \tau_Y \end{bmatrix} = A^{-1}B, \tag{34}$$

where:

$$A = \begin{bmatrix} 1 & \frac{J_x}{J_y} \cdot \tan\theta \sin\phi & -\frac{4J_x \cdot Q_{max} \cdot \tan\theta \cos\phi}{2\sqrt{2}J_z \cdot l \cdot T_{max}} \\ 0 & 1 & \frac{4J_y \cdot Q_{max} \cdot \tan\phi}{2\sqrt{2}J_z \cdot l \cdot T_{max}} \\ 0 & -\frac{2\sqrt{2}J_z \cdot l \cdot T_{max} \cdot \tan\phi}{4J_y \cdot Q_{max}} & 1 \end{bmatrix} \tag{35}$$

$$B = \begin{bmatrix} k_5 \cdot sat(\frac{s_\phi}{\epsilon}) + k_6 \cdot \dot{e}_\phi + \frac{J_x}{2\sqrt{2} \cdot l \cdot T_{max}} \cdot (f_\phi - de\ddot{s}\_\phi) \\ \frac{1}{\cos\phi} \cdot k_5 \cdot sat(\frac{s_\theta}{\epsilon}) + \frac{k_6}{\cos\phi} \cdot \dot{e}_\theta + \frac{J_y}{\cos\phi \cdot 2\sqrt{2} \cdot l \cdot T_{max}} \cdot (f_\theta - de\ddot{s}\_\theta) \\ \frac{\cos\theta}{\cos\phi} \cdot -k_7 \cdot sat(\frac{s_\psi}{\epsilon}) - k_8 \cdot \frac{\cos\theta}{\cos\phi} \cdot \dot{e}_\psi - \frac{J_z \cdot \cos\theta}{4 \cdot Q_{max} \cdot \cos\phi} \cdot (f_\psi - de\ddot{s}\_\psi) \end{bmatrix}. \tag{36}$$

In other words, we have a unique closed-form solution for $\tau_R$, $\tau_P$, and $\tau_Y$ based on all known and measured quantities. Combining $\tau_R$, $\tau_P$, and $\tau_Y$ with $\tau_T$ in (16), we can construct, based on the mixer, the PWM signals to be sent to each of the UAV motors.

Figure 8 shows how $e$ and $\dot{e}$ approach the sliding surface in the yaw motion in one of the simulation tests. Notice that, because of sensor noise, the state of the system cannot come back to (0,0) perfectly. Therefore, the state will move around the origin with a relatively small error due to noise.

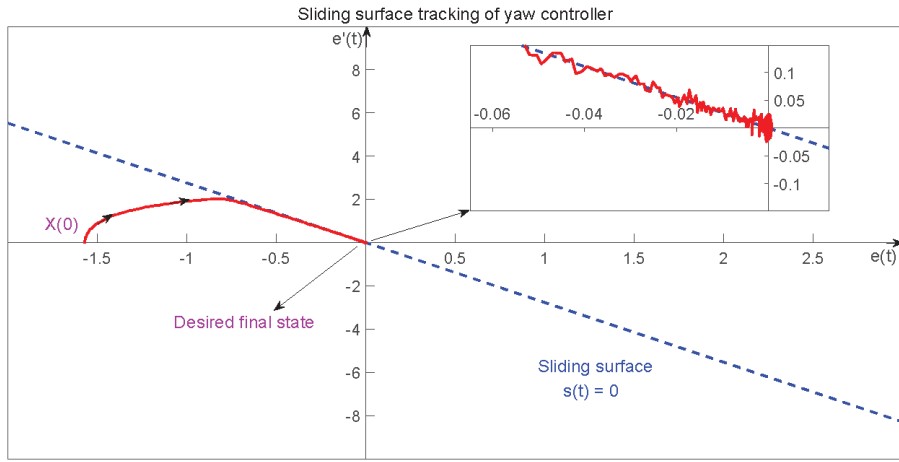

**Figure 8.** Sliding-surface tracking of $\psi$ yaw controller on the plane of $e$ vs. $\dot{e}$.

*2.3. Disturbance Observer*

The SMC controller includes six separate disturbance observers, one for each of X, Y, Z, Yaw, Pitch, and Roll (see Figure 5). In the X, Y, and Z directions, disturbance observers estimate the horizontal and vertical forces, while in the Yaw, Pitch, and Roll directions, disturbance observers estimate the disturbance torque around each axis. The estimated disturbance will then be fed back into the loop and thus be compensated by the SMC controller [16], see Figure 9.

We present below the disturbance observers of x and pitch controllers as examples, since the disturbance observers for the other four directions are quite similar to these two examples. Firstly, for the x controller:

$$\ddot{x} = -\frac{1}{m}[(\sin\phi\sin\psi + \cos\phi\sin\theta\cos\psi) \cdot \boldsymbol{Thrust} + K_1 \cdot \dot{x} \cdot |\dot{x}|] + d, \tag{37}$$

where $d$ is the disturbance assumed to be a continuous signal, independent of the UAV states, and $u_1 = \sin\phi\sin\psi + \cos\phi\sin\theta\cos\psi$ is the control signal. Hence, the disturbance observer is formed as:

$$\begin{cases} d_{ob} = z_{in} + k_{ob}\dot{x} \\ \dot{z}_{in} = -k_{ob} \cdot d_{ob} + k_{ob} \cdot \frac{1}{m}[(\sin\phi\sin\psi + \cos\phi\sin\theta\cos\psi) \cdot \boldsymbol{Thrust} + K_1 \cdot \dot{x} \cdot |\dot{x}|], \end{cases} \tag{38}$$

where $d_{ob}$ is the value, which the observer shows, and $z_{in}$ is an intermediary variable. So, the error between the actual disturbance and the value of the disturbance observers $e^* = d - d_{ob}$ reads:

$$\begin{aligned} \dot{e}^* &= \dot{d} - \dot{d}_{ob} = \dot{d} - \dot{z}_{in} - k_{ob}\ddot{x} \\ &= \dot{d} + k_{ob} \cdot d_{ob} - k_{ob} \cdot \frac{1}{m}[(\sin\phi\sin\psi + \cos\phi\sin\theta\cos\psi) \cdot \boldsymbol{Thrust} + K_1 \cdot \dot{x} \cdot |\dot{x}|] \\ &\quad + k_{ob} \cdot \frac{1}{m}[(\sin\phi\sin\psi + \cos\phi\sin\theta\cos\psi) \cdot \boldsymbol{Thrust} + K_1 \cdot \dot{x} \cdot |\dot{x}|] - k_{ob} \cdot d \\ &= -k_{ob} \cdot e^* + \dot{d}. \end{aligned} \tag{39}$$

Therefore, if $\dot{d}$ is bounded, then the disturbance error $e^*$ is bounded as along $k_{ob} > 0$. Moreover, when $\dot{d} \to 0$, we have $e^* \to 0$. Next, to eliminate the steady-state error caused by disturbances, we should bring the $d_{ob}$ back to the controller. Therefore, to eliminate the influence of disturbance, we bring back the value of the observed disturbance and obtain the new control law:

$$u_{1new} = u_{1original} + d_{ob} \cdot \frac{m}{\boldsymbol{Thrust}}. \tag{40}$$

Similarly, for the pitch controller:

$$\dot{q} = \frac{1}{J_y}[-2\sqrt{2} \cdot T_{max} \cdot l \cdot \tau_P - p \cdot (J_x - J_z) - K_2 \cdot q \cdot |q|] + d, \tag{41}$$

where $d$ is the generic notation for disturbance and should be considered different from that in (37). We have the disturbance observer as follows:

$$\begin{cases} d_{ob} = z_{in} + k_{ob} \cdot q \\ \dot{z}_{in} = -k_{ob} \cdot d_{ob} - k_{ob} \cdot \frac{1}{J_y}[-2\sqrt{2} \cdot T_{max} \cdot l \cdot \tau_P - p \cdot r(J_x - J_z) - K_2 \cdot q \cdot |q|]. \end{cases} \tag{42}$$

Hence, the new control law reads:

$$\tau_{Pnew} = \tau_{Poriginal} + d_{ob} \cdot \frac{J_y}{2\sqrt{2} \cdot T_{max} \cdot l}, \tag{43}$$

where we select $k_{ob} = 5$.

### 2.4. Particle Swarm Optimization for PIDs and SMC

Notice that, in Section 2.2, we established the theoretical conditions guaranteeing the stability of the SMC-drone dynamics based on the Lyapunov stability theory. The next question is then how to optimize the SMC parameters for improved drone performance without violating stability. Since stability conditions are established in Section 2.2, these conditions can guide us to carefully tune the SMC controller gains. To this end, we propose to utilize PSO, as we explain next.

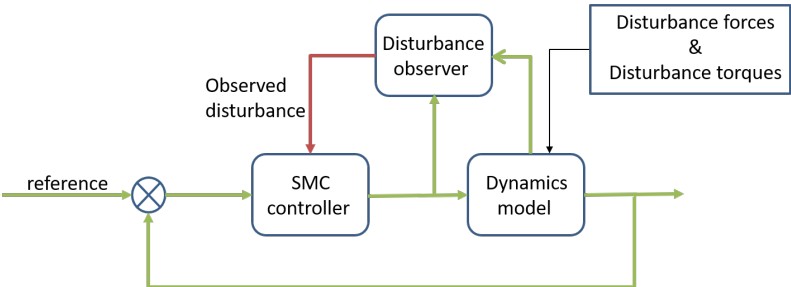

**Figure 9.** Disturbance observer structure combined with SMC controller.

#### 2.4.1. PSO Optimization Procedures

Particle-Swarm Optimization (PSO) is an evolutionary computational method that originated from the study and simulation of birds' predatory behavior [35]. PSO is inspired by the fact that we can imagine birds finding food by using a set of particles to solve a problem. Based on the observation of the activity behavior of a group of birds, the PSO algorithm uses the information-sharing property among individuals in a group to change the movement of the whole group in the problem-solving space from disorder to order, so as to obtain the optimal solution. In addition, compared with other swarm-intelligence algorithm methods [27] and back propagation neural network algorithms [36], PSO has the advantages of fewer parameters to be adjusted, fast convergence, and simple algorithmic structure [37–39].

In detail, the basic idea of PSO is to consider a swarm of particles as a population of potential solutions to a problem. According to some simple mathematical formulations, these particles can move in the problem-solving space to search for solutions. The movement of each particle is guided by its individual best-known position and the best-known position of the entire population of particles (which is also called the global best position). In addition, the individual best position and the global best position should be updated when the particles find other better positions. Thus, when this process is repeated, the whole swarm is expected to move toward a solution of higher quality.

In view of the above, the mathematical expressions for velocity and position updates are given by:

$$\begin{cases} v_{ij}^{k+1} = \omega v_{ij}^{k} + c_1 r_1 \left( Pbest_{ij} - x_{ij}^{k} \right) + c_2 r_2 \left( Gbest_{j} - x_{ij}^{k} \right) \\ x_{ij}^{k+1} = x_{ij}^{k} + lr \cdot v_{ij}^{k+1}, \end{cases} \tag{44}$$

where $1 \leq i \leq N$, $1 \leq j \leq D$, $1 \leq k \leq K$. Here, $N$ is the number of particles in the swarm, $D$ is the dimension of particle (the number of the parameters), $K$ is the maximum number of iterations, $x_{ij}^{k+1}$ is the position of particle $i$ in the dimension $j$ at iteration $k$, $v_{ij}^{k+1}$ is the velocity of particle $i$ in the dimension $j$ at iteration $k$, $Pbest_{ij}$ is a personal best position of particle $i$ in the dimension $j$, $Gbest_{ij}$ is a global best position of all particles in the dimension $j$, $\omega$ is the inertia weight factor, $c_1$ and $c_2$ are the acceleration constants, $r_1$ and $r_1$ are random numbers in the interval [0, 1], and $lr$ is the learn rate ($0 \leq lr \leq 1$), which is the proportion at which the velocity affects the movement of the particle (where $lr = 0$ means velocity will not affect the particle at all, and $lr = 1$ means velocity will fully affect the particle) [37].

To choose the *Pbest* and *Gbest* of the particles, we use a scoring system concerning three standards: the Integraltime absolute error (ITAE), settling time (Ts), and the absolute

value addition of overshoot and undershoot between desired trajectory and actual flight path ($OsUs = abs(overshoot) + abs(undershoot)$). As the performance of the controller is superior when all these three standards have smaller values, and we want the change of scores be more sensitive when the controller has a satisfactory performance, we use the normal distribution to build the score model. Moreover, we divide the parameters into three parts: to control the x and y motion, to control the z motion, and to control the yaw angle. Each part will have their own scoring system, and the general formula is as follows:

$$Score_i = 100 \cdot \frac{1}{\sigma_i \sqrt{2\pi}} e^{\frac{-(x_i)^2}{2\sigma_i^2}}, \qquad (45)$$

where $Score_i$ is the score when the standard has $x_i$ value, where $x_i$ is either ITAE, Ts, or OsUs, and the value of $\sigma_i$ is shown in Table 6. Thus, we have the scores shown in Figure 10.

**Table 6.** Value of $\sigma$.

| i | $ITAE_{x+y}$ | $ITAE_z$ | $ITAE_{yaw}$ | $OsUs_{x+y}$ | $OsUs_z$ | $OsUs_{yaw}$ | $Ts_{x+y}$ | $Ts_z$ | $Ts_{yaw}$ |
|---|---|---|---|---|---|---|---|---|---|
| $\sigma_i$ | 18 | 36 | 3 | 15 | 0.5 | 15 | 25 | 4 | 19 |

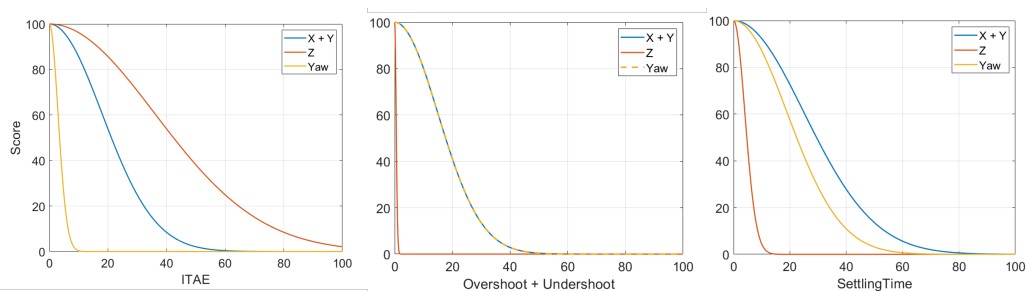

**Figure 10.** (**Left**): Defined ITAE score curve for X, Y, Z, and YAW; (**Middle**): Integrated Overshoot and Undershoot (OsUs) score curve for X, Y, Z, and YAW; (**Right**): Defined Settling Time (Ts) score curve for X, Y, Z, and YAW.

The flow chart of the PSO algorithm is shown in Figure 11. In this paper, the parameter initialization process is omitted, but the core idea is to initialize the parameters within a reasonable range of the SMC parameters respecting the stability criterion, and the initialization script will select the initial particle population. Then, the flight performance is scored according to the simulation of the mathematical UAV model. Based on the scoring results, the algorithm can find the individual best position of each particle and the best-known position of the swarm of all particles; thus, the position and velocity of the particles can be updated and iterated based on the mathematical formula. Finally, after a sufficient number of iterations, we can arrive at a final solution to the problem.

For the case of SMC, optimization can iterate while respecting the theoretical stability conditions derived in Section 2.2 on the SMC parameters. In other words, the SMC stability conditions can be incorporated into PSO to assure that PSO always suggests a stabilizing set of SMC parameters. On the other hand, since we do not have explicit stability conditions for the PID controllers, there is no guarantee that the PSO at each iteration will yield stabilizing parameters. However, we judiciously select the initial PID gains in view of PX4 documentation and find out that PSO eventually yields optimized and stabilizing PID gains.

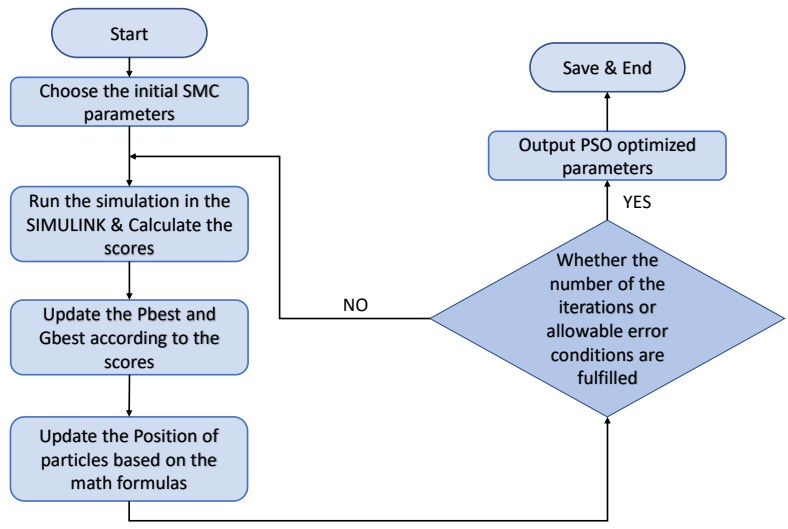

**Figure 11.** Process flow of the PSO.

### 2.4.2. PSO-Tuned Results

In the simulations, we choose:

- The initial state: $x_0 = 0\ m$, $y_0 = 0\ m$, $z_0 = 0\ m$, $\psi_0 = 0\ degrees$;
- The desired position and yaw angle: $x_d = 4\ m$, $y_d = 4\ m$, $z_0 = 20\ m$, $\psi_0 = 90\ degrees$;
- The PSO algorithm parameters: $\omega = 0.4$, $c_1 = 0.5$, $c_2 = 0.5$, $lr = 0.5$.

During the PSO tuning, we observed the change of scores and values of the three predefined standards in Figure 12.

From Figure 12, the total score keeps increasing, even though one of the criteria fluctuates up and down, and this swarm of particles almost finds the optimal solution at around 20 iterations. Finally, after 100 iterations, we obtain a high total score and small ITAE, short settling time, and low overshoot and undershoot, which means expected flight performance. The optimized parameters for SMC controller are provided in Table 7, and the fixed parameters for SMC are listed as following: $\epsilon = 0.5$, $k_1 = 0.001$, $k_3 = 0.1$, $k_{ob} = 5$.

**Table 7.** SMC Controller parameters after PSO Tuning.

| Parameter | $a_1$ | $a_2$ | $k_2$ | $b_{1x}$ | $b_{2x}$ | $b_{1y}$ | $b_{2y}$ | $c_x$ | $c_y$ |
|---|---|---|---|---|---|---|---|---|---|
| Value | 3.5524 | 10.0000 | 3.5524 | 0.7426 | 1.4305 | 0.5930 | 1.0895 | 0.2871 | 0.3867 |
| | $b_3$ | $b_4$ | $a_3$ | $k_5$ | $k_6$ | $a_4$ | $k_7$ | $k_8$ | $k_4$ |
| | 0.6373 | 0.0223 | 8.1910 | 0.0388 | 0.0219 | 2.7640 | 0.3147 | 0.1244 | 0.5000 |

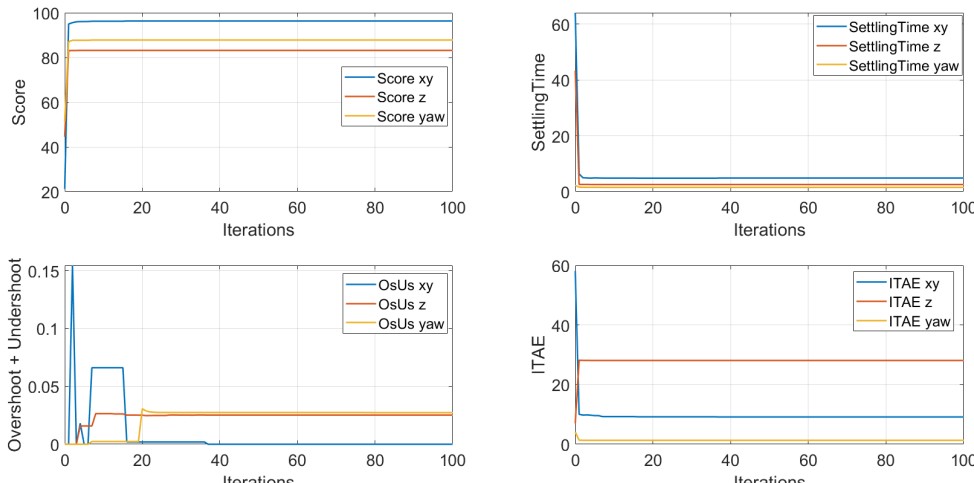

**Figure 12.** Score and standards vs. PSO Iterations. (**Top-left**): Weighted average score of Ts, OsUs, and ITAE combined for each direction; (**Top-right**): Score of Ts for each direction; (**Bottom-left**): Score of OsUs for each direction; (**Bottom-right**): Score of ITAE for each direction.

It is important to note that the above PSO algorithm is run offline, meaning it is iterated first in conjunction with the SIMULINK model, to reveal the optimum control parameters. Once these parameters are revealed (Table 7), we can then implement them in SIMULINK as well as in PX4 for comparison purposes. In Figure 13, we compare the performance of the initialized and optimized SMC parameters and observe an improvement in the controller performance. In the left panel of Figure 13, the settling time of XYZ is significantly reduced, and, in the right panel of Figure 13, the pitch and roll change rapidly and the yaw angle converges quickly to the reference value.

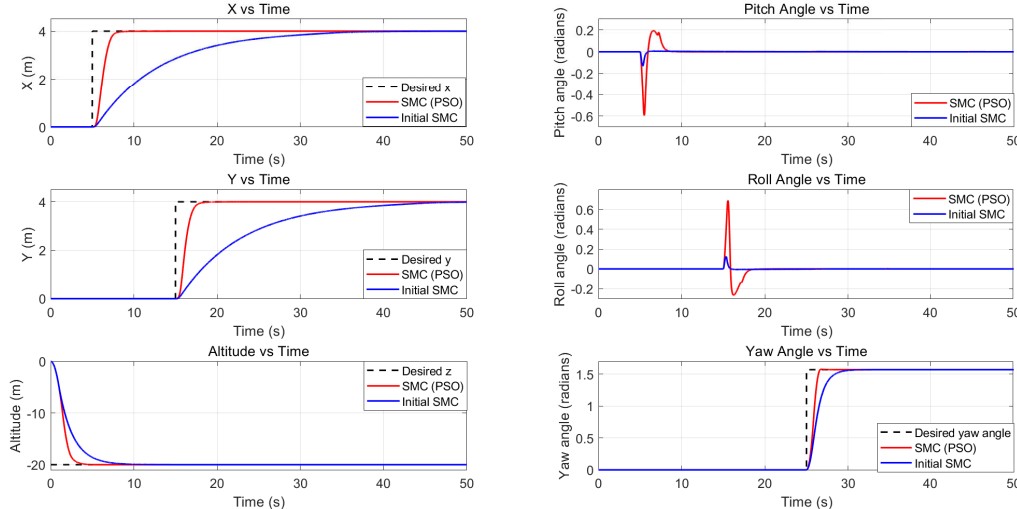

**Figure 13.** Comparison of step-tracking performance before and after the PSO tuning. Simulations are based on the SIMULINK mathematical UAV model.

As a control group, we also need to use the same PSO method to adjust the PID position and PID rate parameters. For the sake of brevity, here we provide only the results of PSO, see Tables 8 and 9.

**Table 8.** PSO-tuned parameters in PID-position-only controller.

| PID Gain | x | y | z | Yaw | Pitch | Roll |
|---|---|---|---|---|---|---|
| Kp | 25.6353 | 27.1769 | 3.2393 | 0.3017 | 0.0885 | 0.0885 |
| Ki | 0.1436 | 0.2372 | 0.2637 | 0.0004 | 0.0113 | 0.0113 |
| Kd | 21.3759 | 20.2633 | 2.2944 | 0.1879 | 0.0398 | 0.0398 |

**Table 9.** PSO-tuned parameters in PID-rate controller.

| PID Gain | x | y | z | vx | vy | vz | Yaw | Pitch | Roll |
|---|---|---|---|---|---|---|---|---|---|
| Kp | 1.5551 | 1.7964 | 5.5146 | 28.5256 | 27.4139 | 5.5394 | 0.2478 | 0.0922 | 0.0922 |
| Ki | 0 | 0 | 0.0617 | 0 | 0 | 0.2710 | 0.0025 | 0.0141 | 0.0141 |
| Kd | 0.2287 | 0.9217 | 1.8326 | 0.1133 | 0.1745 | 0.3772 | 0.1689 | 0.0476 | 0.0476 |

## 3. Simulation Results

Now that we have PSO-optimized SMC and PSO-optimized PID controllers at hand, we next incorporate these controllers into PX4 under various scenarios. We note here that PSO is implemented offline a priori. In PX4 simulations, we do not perform PSO optimization in real time.

### 3.1. Trajectory Tracking Performance

In the previous sections, the SMC controller was designed on a fully nonlinear mathematical model of a quadrotor aircraft. In this section, the SMC controller has been continuously developed under the PX4 architecture and validated with a PX4-simulated UAV, which means that environmental noise and uncertainties have been taken into account. Several test scenarios, including step tracking, slope tracking, and force and torque disturbances have been applied. All small oscillations in position and attitude are considered to be the combined result of sensor noise, wind interference, discretization, time delay, and vehicle dynamics uncertainty. As a baseline for comparison, a PSO-tuned PID-position controller and a PSO-tuned PID-rate were also tested.

The following is the list of standards used to evaluate the performance of the controllers:

- The Maximum Overshoot (Mp);
- The Settling time (Ts);
- The Rise time (Tr);
- The Integral absolute error (IAE);
- The Integral time absolute error (ITAE);
- The Integral square error (ISE);
- The Integral time squared error (ITSE).

The IAE, ITAE, ISE, and ITSE are assumed to be measured starting from two times the settling time ($2 \cdot Ts$) and last for 10 s ($2 \cdot Ts + 10$), during which we assume that the UAV reached a relatively stable state.

We would like to find out which controller has less overshoot, less settling time, and smaller IAE, ITAE, ISE, and ITSE. However, rise time is not our main concern, as it can easily lead to large overshoot and significant wander of the UAV. From our point of view, the desired controller should enable smooth, rapid, and accurate UAV movement.

### 3.1.1. Step Profile Tracking

In the first case of this study, the UAV is controlled to track a step contour in terms of distance, height, and angle, respectively. Throughout the time axis, the UAV should rise from 0 m to 20 m, starting at 5 s and moving from 0 m to 4 m in the X and Y directions, starting at 15 s, and rotating 90 degrees starting at 40 s, as shown in the black dash reference curves in Figure 14.

Table 10 summarizes the simulation results, which show the SMC controller has the smallest overshoot and the smallest settling time. Compared to the PID controllers, its rise

time becomes slightly longer in the horizontal position (X and Y), but slightly shorter in height and angle (Z and YAW). Furthermore, we analyzed the zoom-in figures of z and yaw motion. The PID-based controllers need a longer time to reach steady states, which leads to a small steady-state error even after a long period of time. In contrast, the SMC UAV does not have many errors, given its IAE, ITAE, ISE, and ITSE values are relatively small. Figure 14 shows that SMC motion has a smaller range and fewer oscillations, and its time derivative is smaller, compared to the PID, especially in terms of pitch and roll tracking. If we assume that smoothness and stability have greater weights in our evaluation, SMC is more preferable.

**Table 10.** Step-tracking performance.

|  | Controller | Mp | Ts | Tr | IAE | ITAE | ISE | ITSE |
|---|---|---|---|---|---|---|---|---|
| X | $PID_{POS}$ | 52.0197% | 4.6427s | 0.7134s | 0.2610 | 14.2687 | 0.0077 | 0.4232 |
|  | $PID_{RATE}$ | 14.0381% | 3.1368s | 0.8642s | 0.1677 | 8.4630 | 0.0037 | 0.1854 |
|  | SMC | 4.5037% | 2.2240s | 1.3705s | 0.1773 | 8.9542 | 0.0038 | 0.1915 |
| Y | $PID_{POS}$ | 6.6901% | 2.6134s | 0.8304s | 0.2556 | 12.6073 | 0.0074 | 0.3599 |
|  | $PID_{RATE}$ | 7.4420% | 3.7456s | 0.8379s | 0.1305 | 6.7214 | 0.0025 | 0.1258 |
|  | SMC | 3.8777% | 2.3412s | 1.4672s | 0.1096 | 5.2685 | 0.0017 | 0.0823 |
| Z | $PID_{POS}$ | 15.0943% | 3.9990s | 1.5232s | 0.8560 | 27.4586 | 0.0817 | 2.5453 |
|  | $PID_{RATE}$ | 6.0805% | 3.4192s | 1.7296s | 0.2363 | 7.1725 | 0.0077 | 0.2246 |
|  | SMC | 3.0367% | 2.4962s | 1.6328s | 0.0606 | 1.8577 | 0.0006 | 0.0174 |
| Yaw | $PID_{POS}$ | 2.6836% | 1.5183s | 1.0185s | 0.9969 | 62.8274 | 0.1010 | 6.3625 |
|  | $PID_{RATE}$ | 2.7375% | 1.6036s | 1.0796s | 6.4843 | 409.2521 | 4.2104 | 265.3702 |
|  | SMC | 0.0390% | 1.4370s | 0.9618s | 0.0951 | 5.9757 | 0.0013 | 0.0838 |

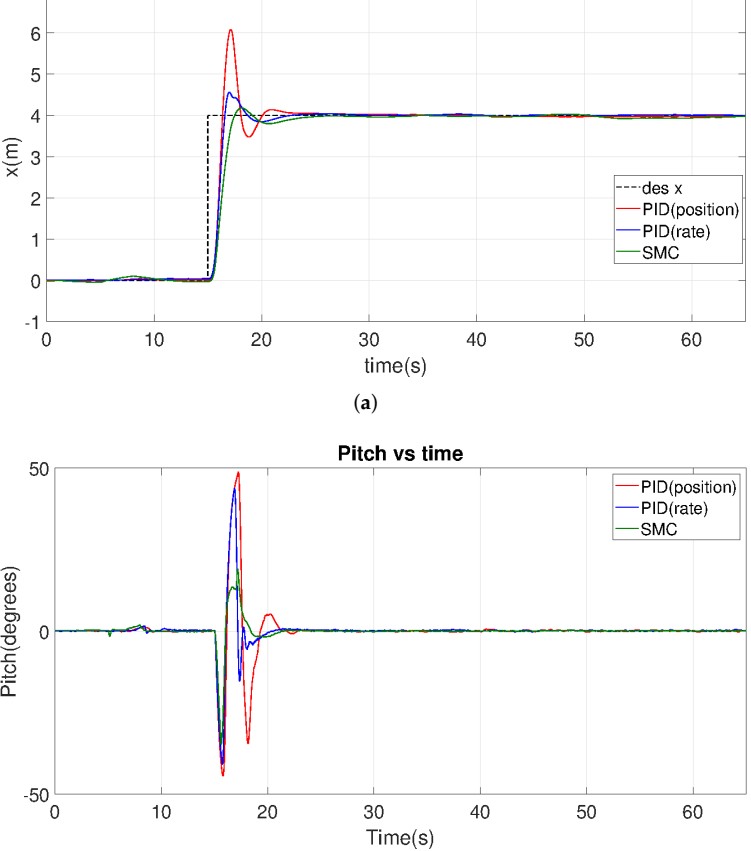

**Figure 14.** *Cont.*

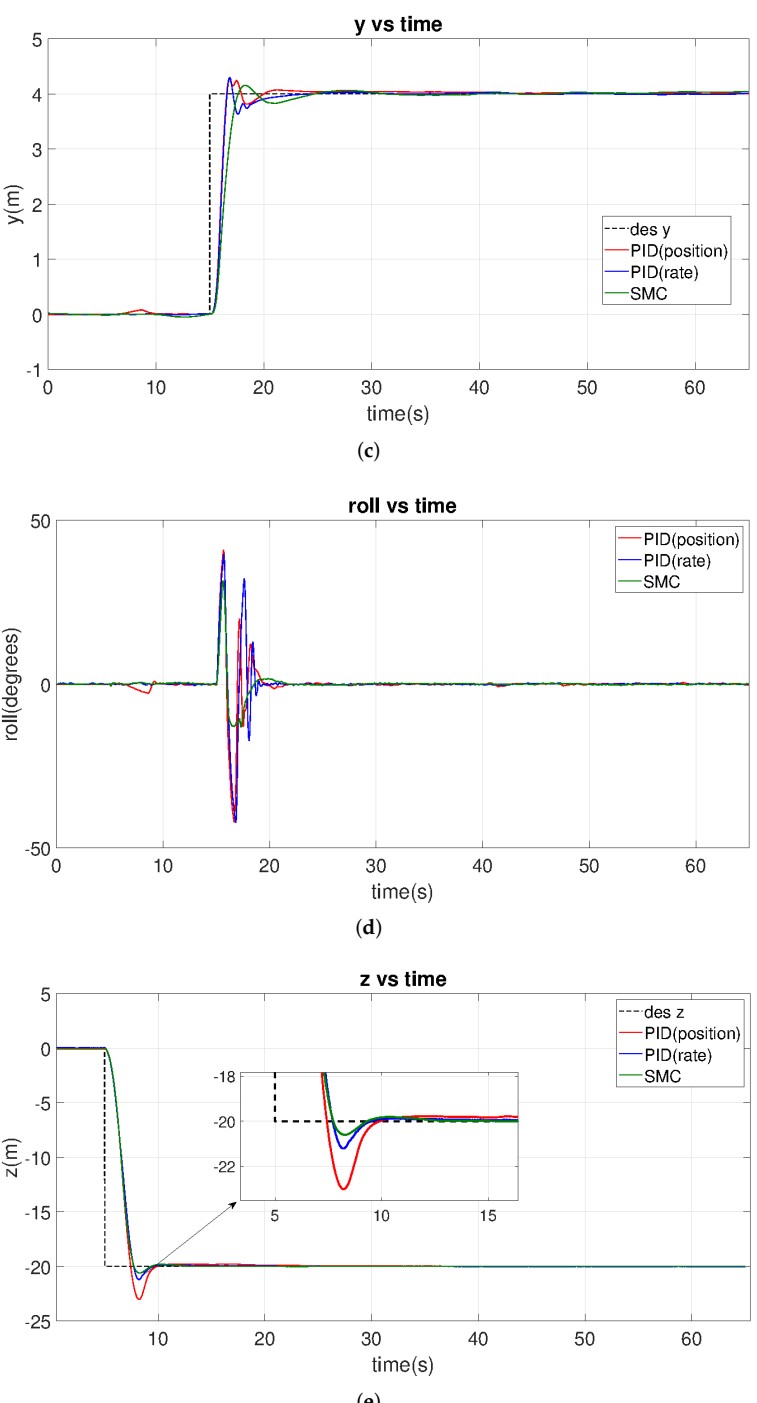

**Figure 14.** *Cont.*

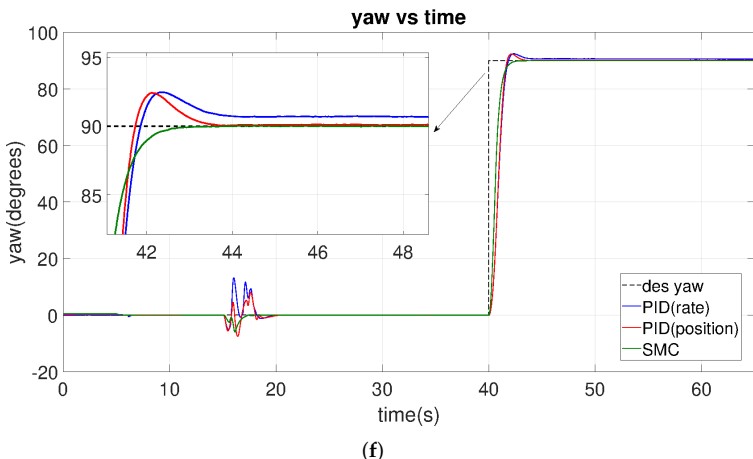

**Figure 14.** Comparison of step-tracking performance of PSO optimized SMC, PID-position, and PID-rate controllers from the PX4 conducted jMavsim simulation under normal noise levels. (**a**) X vs. Time. (**b**) Pitch vs. Time. (**c**) Y vs. Time. (**d**) Roll vs. Time. (**e**) Z vs. Time. (**f**) Yaw vs. Time.

It can be observed that the proposed SMC outperforms the other two PID control methods in terms of tracking accuracy. The benchmark PID-position and PID-rate controllers can stabilize the simulated UAV under PX4 architecture but with a larger overshoot. These phenomena are undesirable in reality, especially when tracking accuracy and stability are required. On the other hand, the proposed SMC can exhibit much improved tracking performance.

### 3.1.2. Ramp Profile Tracking

The second case focuses on the ramp tracking performance. This is also important when the UAV is intended to perform curvilinear trajectory tracking. We separate the ramp experiment into four discrete scenarios: (a) ramp only in the X direction; (b) ramp only in the Y direction; (c) ramp only in the Z direction; and (d) ramp only in the Yaw.

The tracking results for this case study are given in Figure 15, where the error histograms in panels b, d, f, and h show that the PID-rate-based controller design exhibits much larger tracking error (blue bars shift away from the origin in the error histograms). Although the PID-position controller gives a finer tracking performance than the PID-rate, see panels a, c, e, and g, and behaves similarly to the SMC, it is more oscillatory.

In detail, for X and Y slope tracking, SMC fits the reference value best (compare IAE, ITAE, ISE, and ITSE on Table 10), and the PID-position controller, however, has a relatively smaller tracking error than the PID-rate controller (Table 10); it continuously oscillates around the reference slope (panels a, c), which results in a wider position error distribution around the zero-error than SMC, see panels b and d. A similar result arises in the Z-axis slope tracking, where the green area (SMC) is larger than the red area (PID-position) near the zero-error boundary, see panel f. Finally, the yaw angular tracking performance of SMC outperforms the PID controllers, see panel g zoom-in plot.

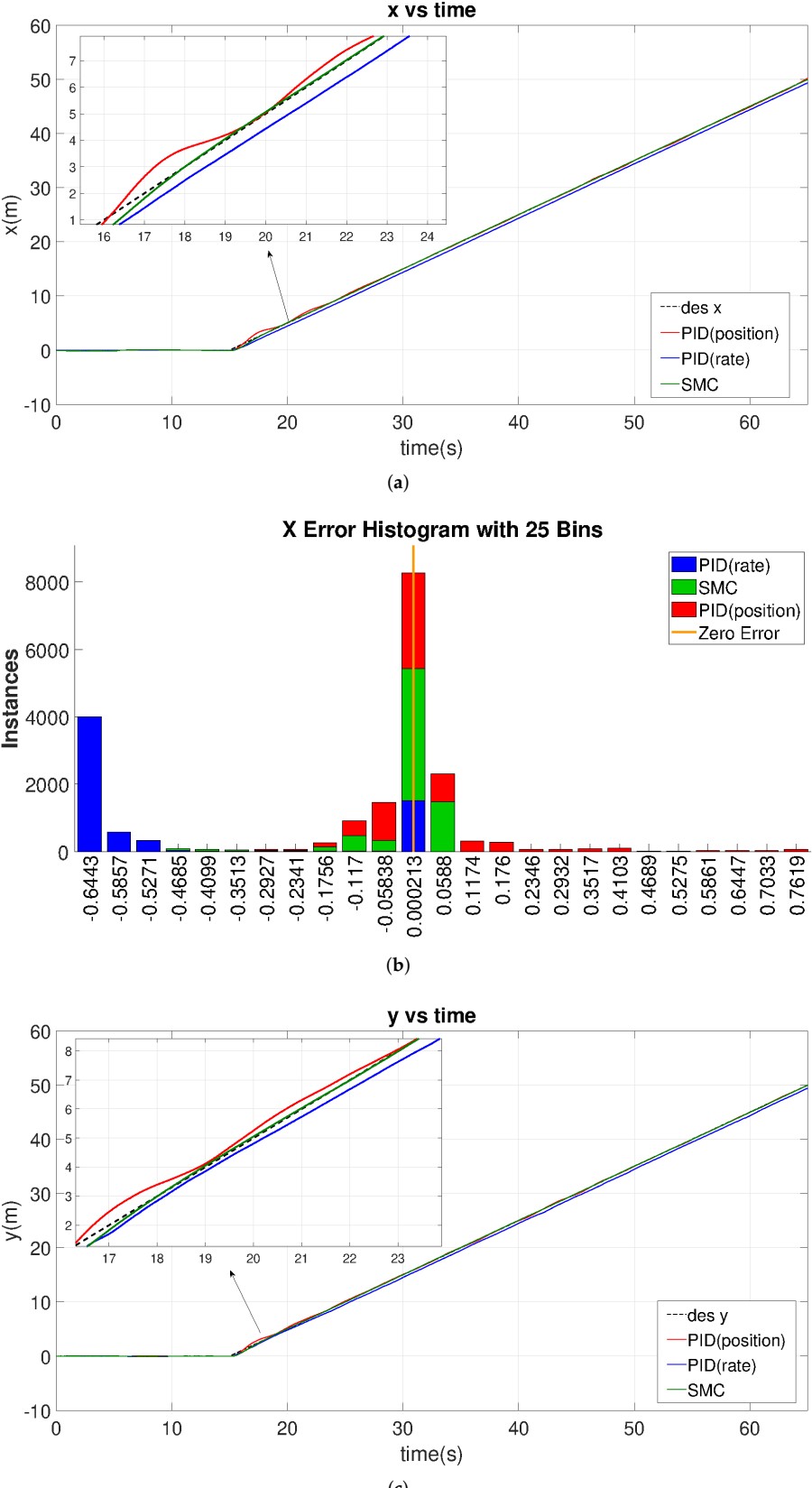

**Figure 15.** *Cont.*

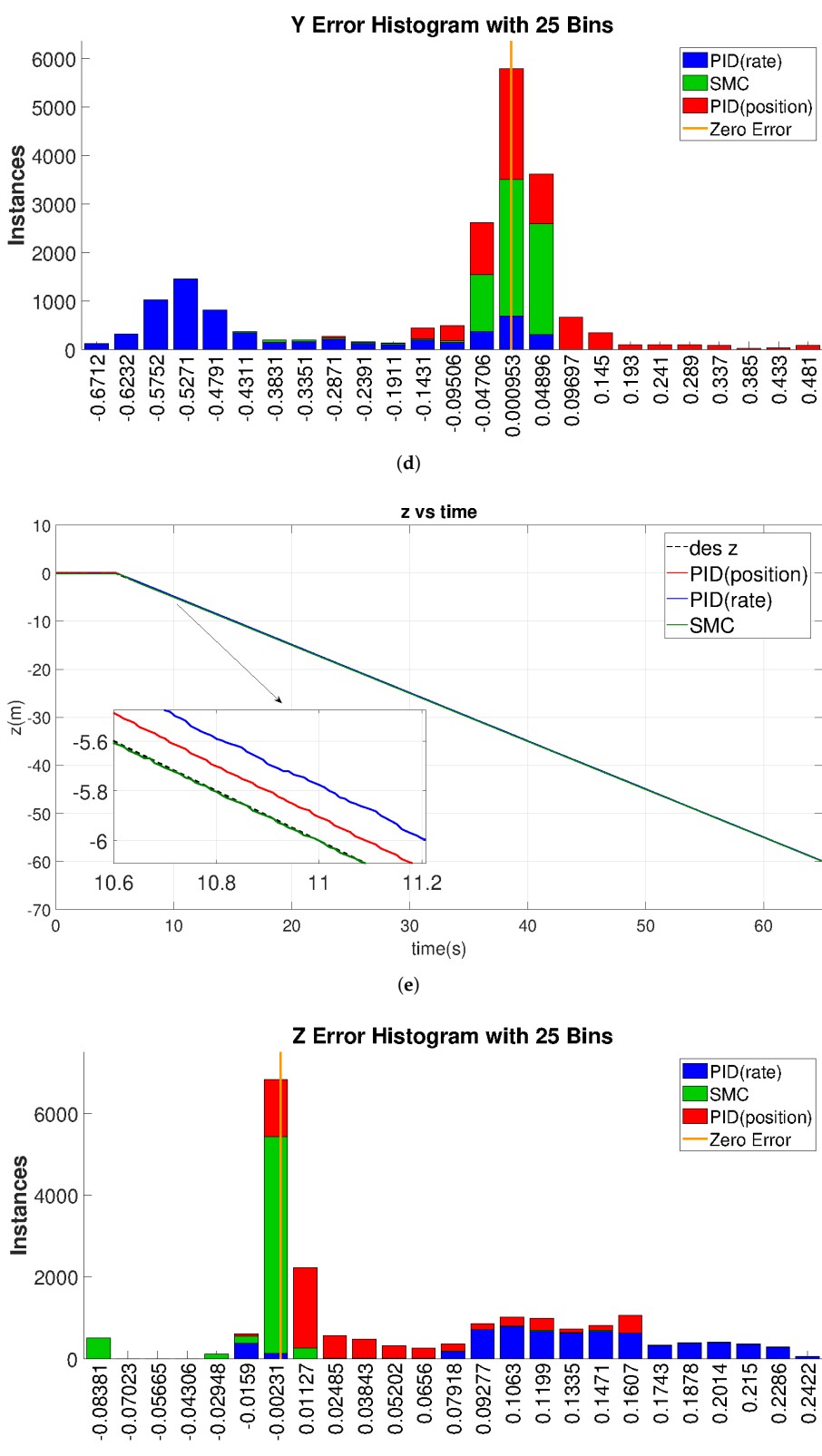

**Figure 15.** *Cont.*

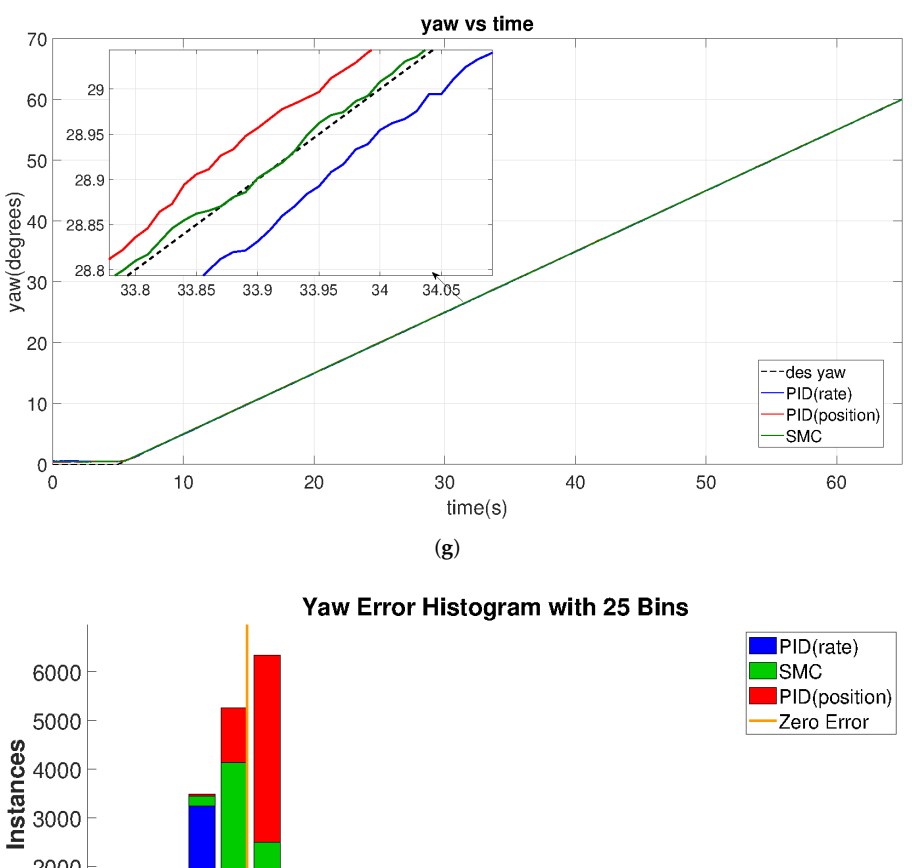

(g)

(h)

**Figure 15.** Comparison of ramp tracking with error histogram from the PX4 conducted jMavsim simulation under normal noise levels. (**a**) X Ramp vs. Time. (**b**) X Error histogram. (**c**) Y Ramp vs. Time. (**d**) Y Error histogram. (**e**) Z Ramp vs. Time. (**f**) Z Error histogram. (**g**) Yaw Ramp vs. Time. (**h**) Yaw Error histogram.

### 3.1.3. Complex Trajectory Tracking

Finally, we introduce a complex trajectory which is a combination of X, Y, and Z position set-points. We set these references as: $x = 10 \cdot \cos(-0.125t + 3.75) \cdot \sin(0.25t - 7.5)$; $y = 10 \cdot \sin(0.125t - 3.75) \cdot \sin(0.25t - 7.5)$; $z = 10$. The UAV takes off at $0s$ and starts tracking the trajectory at $30s$. The result is shown in Figure 16a (top view), from which we can see the trajectory resembles a flower. To compare the overall tracking performance of three controllers, the figure also introduces a timeline in Figure 16b,c.

From the top view of Figure 16a, we observe that SMC tracks the desired trajectory during the simulation, while PID-position- and PID-rate-based controllers show notable tracking errors and deviations continuously. Panel a also shows that PID-position produces oscillations around the reference (overshoots and undershoots). On the other hand, from the time trace of x and y in Figure 16b,c, we observe the PID-rate controller has a persistent time delay (about 1 s) compared with SMC and the desired set-points.

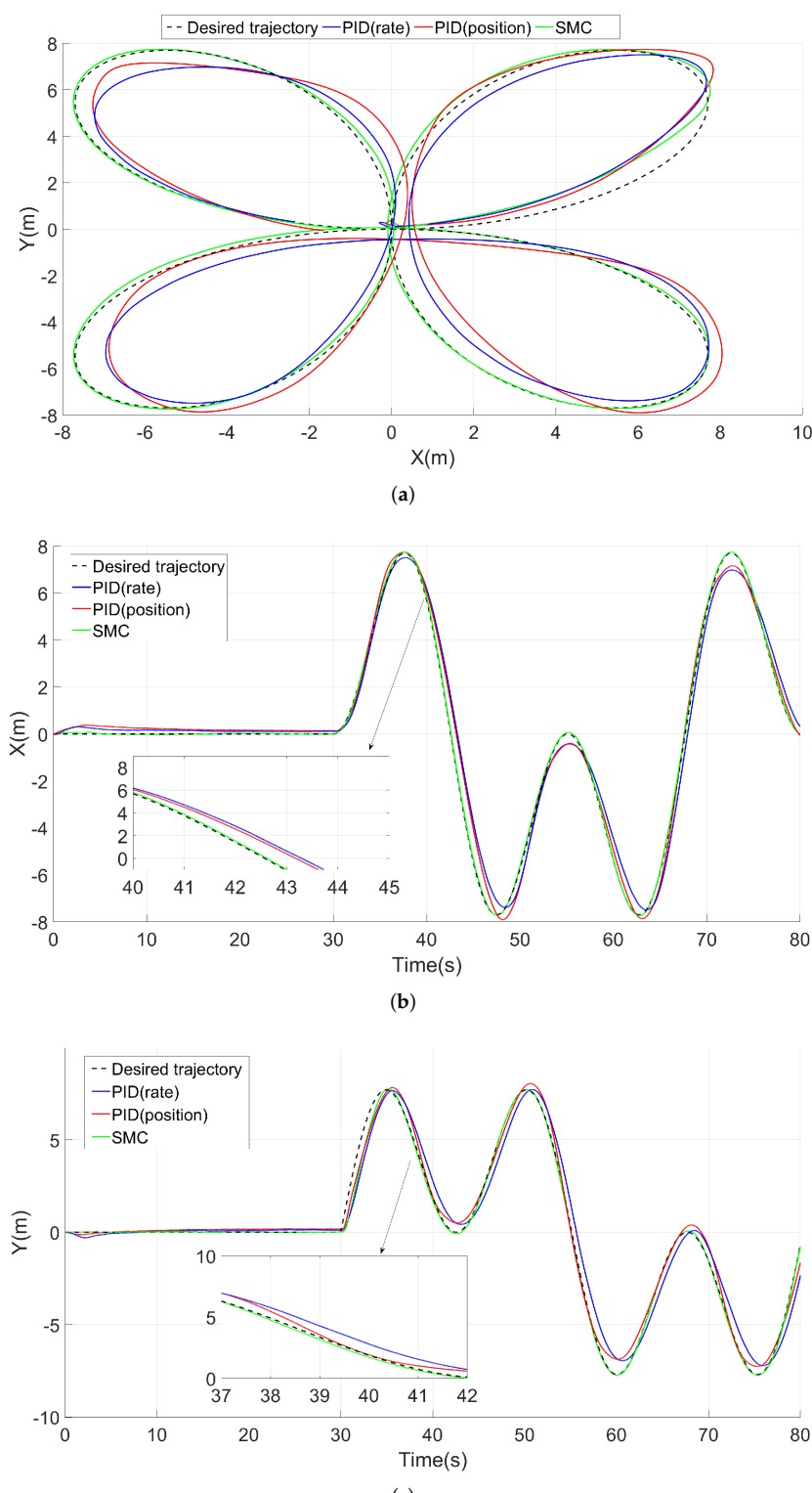

**Figure 16.** Comparison of flower-pattern complex trajectory tracking results from the PX4 conducted jMavsim simulation under normal noise levels. (**a**) Top view of tracking results of a flower-pattern complex trajectory. (**b**) Tracking results in x direction with the timeline. (**c**) Tracking results in y direction with the timeline.

Figures 17 and 18 show the error histograms of the SMC controller and the PID controllers during the flower-pattern trajectory-tracking simulations. These two figures show that the position errors of SMC are mostly concentrated around zero with respect to the both x and y

directions, i.e., −0.19 m to +0.06 m in the x direction and −0.08 m to +0.23 m in the y direction; conversely, the position-error distribution of PID controllers is relatively scattered across the range of errors in the histograms. We can observe that SMC responded quicker than the PID controllers from Figure 16 (see when the slope changes in the desired trajectory), and, meanwhile, it produces smaller position-tracking errors as evident in Figures 17 and 18. Therefore, combining the performances of the tracking accuracy and tracking speed, SMC demonstrates superiority over these two PID controllers with respect to complex trajectory tracking.

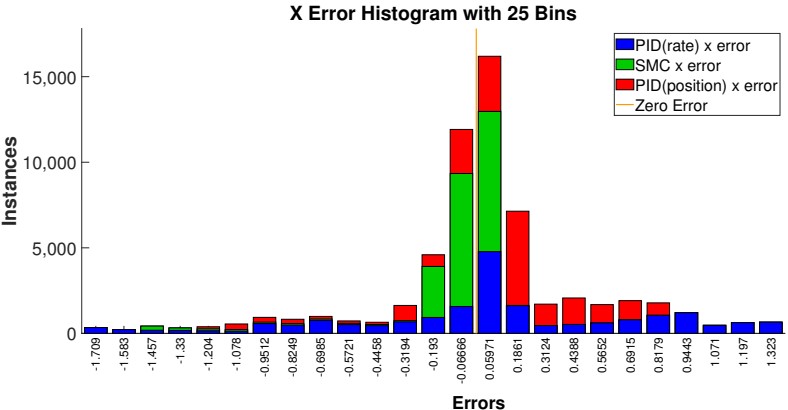

**Figure 17.** X position error histogram tracking the flower-pattern complex trajectory.

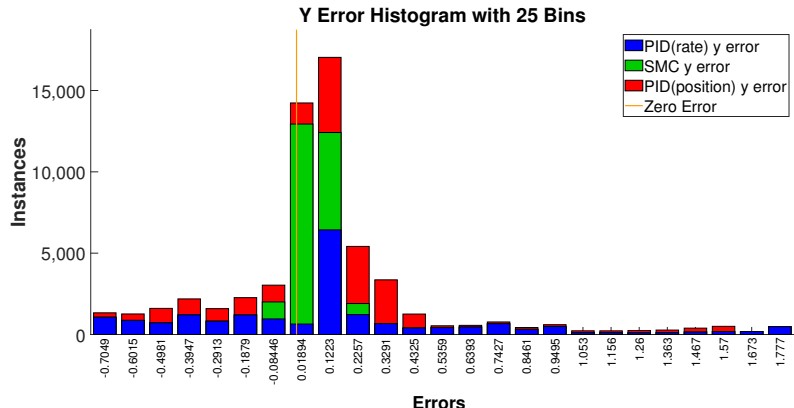

**Figure 18.** Y position error histogram tracking the flower-pattern complex trajectory.

### 3.2. Disturbance Resistance

Accurate control of position and altitude in environments with disturbances is critical to the safety of UAVs. In this study, the UAV was controlled to maintain its position under disturbances in different directions. There are six discrete scenarios: (1) Constant force disturbance in X direction, (2) Constant force disturbance in Y direction, (3) Constant force disturbance in Z direction, (4) Constant torque disturbance around X axis, (5) Constant torque disturbance around Y axis, and (6) Constant torque disturbance around Z axis.

#### 3.2.1. Force Disturbances

The tracking performance, together with their force disturbance estimates from the proposed disturbance observer, is illustrated in Figure 19. In scenario (1), panel (a) and panel (b) correspond to a constant force disturbance added in the X direction as $1\,\text{m/s}^2$; in scenario (2), panel (c) and panel (d) correspond to a constant force disturbance added in the Y direction as $1\,\text{m/s}^2$; and in scenario (3), panel (e) and panel (f) correspond to a constant force disturbance added in the Z direction as $10\,\text{m/s}^2$.

Despite the relatively small gain used in the design of the observer ($k_{ob} = 5$), disturbance estimates can quickly converge to the true value of the disturbance. It can also be observed that the proposed SMC controller outperforms the other two controllers in resisting the force disturbance. In both the vertical and horizontal directions, the PID-position and PID-rate controllers have large oscillations in the presence of disturbance bursts, and they have large steady-state errors in the presence of disturbances. These phenomena are undesirable, especially when the natural wind changes rapidly or when the payload changes, and these characteristics are likely to cause a true UAV crash. On the other hand, SMC control can use the estimated disturbance information to form an active compensation-control effort. As a result, it exhibits a much-improved force-interference-rejection capability with a higher tracking accuracy.

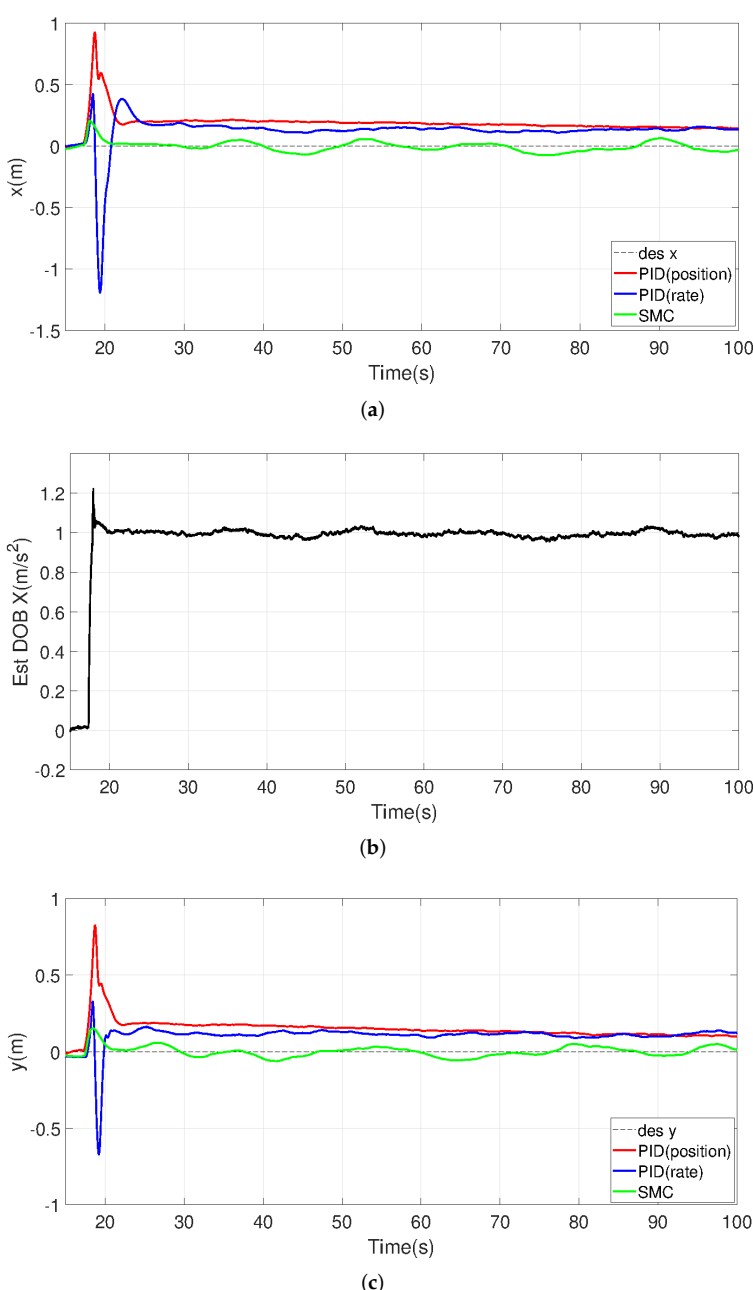

**Figure 19.** *Cont.*

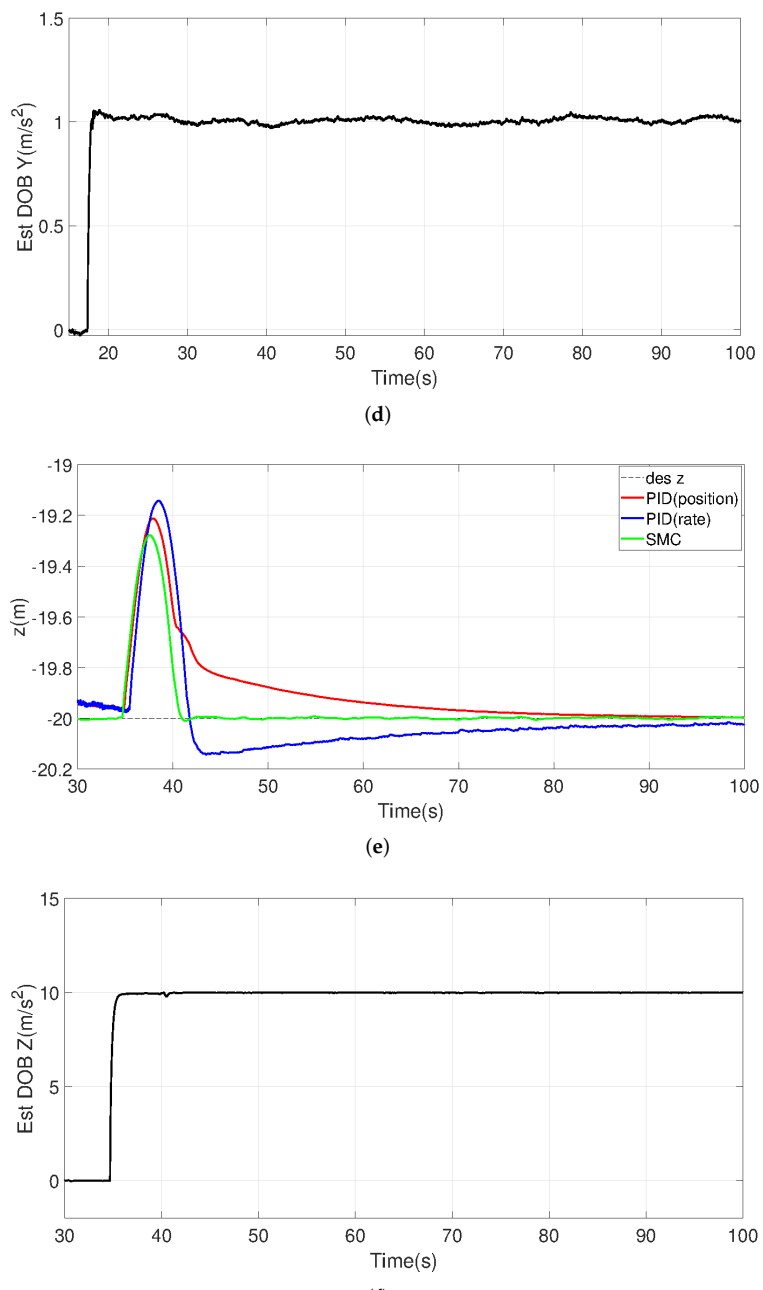

**Figure 19.** Origin-tracking results and disturbance estimation results under constant force-disturbance. (**a**) X vs. Time. (**b**) Estimated disturbance in the x direction. Actual disturbance is $1 \, \text{m/s}^2$. (**c**) Y vs. Time. (**d**) Estimated disturbance in the y direction. (**e**) Actual disturbance is $1 \, \text{m/s}^2$. (**f**) Z vs. Time. Estimated disturbance in the z direction. Actual disturbance is $10 \, \text{m/s}^2$.

3.2.2. Comparison of the Wind Resistance under the Cross-Wind Effect

Crosswind, as a special form of disturbance, often occurs when UAVs are flying, especially in indoor or narrow environments, and these winds can be superimposed in multiple directions and are often unpredictable. Therefore, the wind resistance of UAVs is often used as an important factor to evaluate their flight-control performance. For this reason, in addition to the tracking performance, we investigate the effects of variable crosswinds. The wind resistance results in the horizontal plane for each controller are listed in Figures 20 and 21, with X representing north and Y representing east. The corresponding estimates of the disturbances based on the disturbance observer in SMC are also shown in Figure 20.

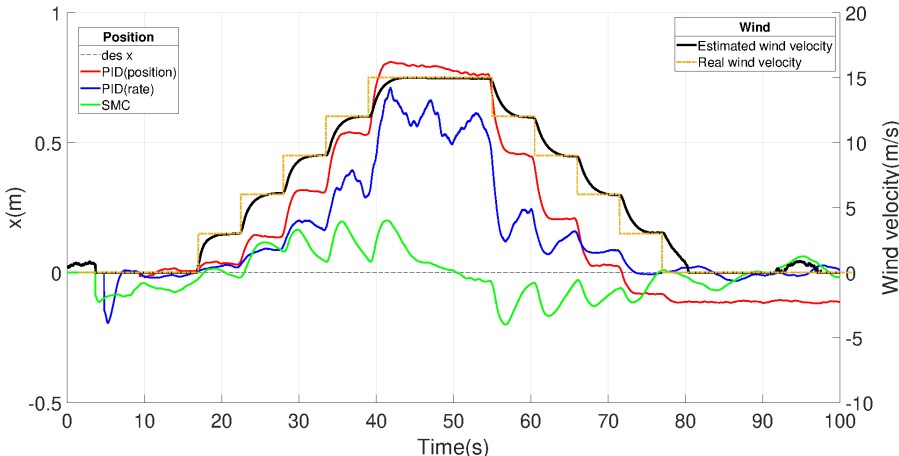

**Figure 20.** Comparison of the origin tracking performance under unidirectional crosswind effect and SMC disturbanceobserver real-time wind velocity estimation results.

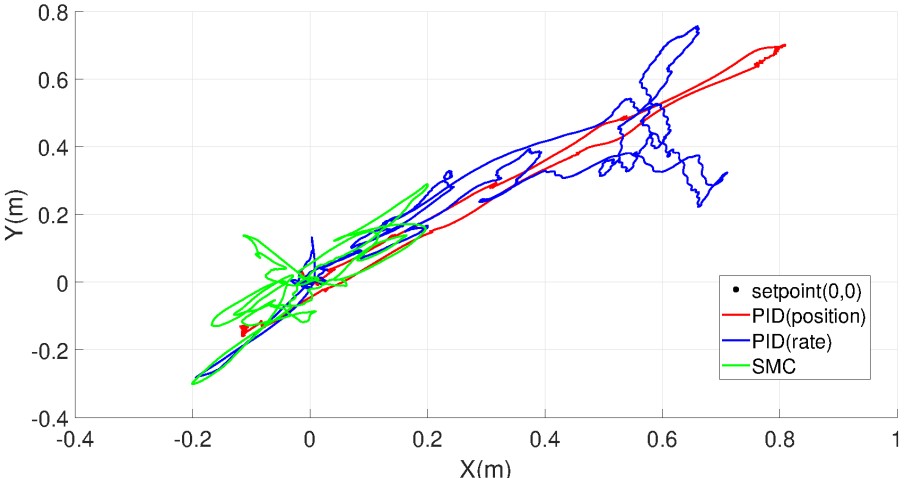

**Figure 21.** Top view of the origin-tracking results under unidirectional crosswind effect.

First, we add a variation of horizontal wind as shown in Figure 20. Starting at 18 s, we add 3 m/s (about 5.83 Kts) of wind in the X and Y in each 5 s, which means the resultant wind added in each period should be $3\sqrt{2}$ m/s in the northeast direction. Although the wind added looks more similar to step functions, the actual wind's increasing rate was around 4.2426 m/s², which means that the wind speed was accelerated to the reference value in about 1 s. Starting from around 55 s, the wind speed was gradually reduced down to 0 m/s. Throughout the simulations, SMC was able to reject wind forces and maintain the desired point within ±20 cm, but the PID drifted considerably, up to 75 cm away from the desired point. During the test, the maximum wind resistance of the SMC- and PID-controlled PX4 UAV was 30.5 m/s (about 59.28 Kts); exceeding this value would result in irreversible trajectory deviation, leading to instability.

In addition, the wind speed estimated by the disturbance observer is also shown, which matches well with the actual wind-speed magnitude. Due to the symmetry of the UAV, the response on the Y-axis is similar to the response on the X-axis, and the Y-plot is omitted here for clarity. It can be observed from Figure 21 that UAV motion with the SMC controller is concentrated more around the origin than with other controllers; hence, SMC has the best crosswind resistance in terms of position accuracy. This is due to its active disturbance compensation.

### 3.2.3. Torque Disturbances

The tracking performance together with the torque disturbance estimates from the proposed disturbance observer is illustrated in Figure 22. In scenario (4), panels (a) and (b) correspond to a constant torque disturbance added around the X axis as $5 \text{ rad/s}^2$; in scenario (5), panels (c) and (d) correspond to a constant torque disturbance added around the Y axis as $5 \text{ rad/s}^2$; in scenario (6), panels (e) and (f) correspond to a constant torque disturbance added around the Z axis as $1 \text{ rad/s}^2$.

It can be seen that the proposed SMC controller is superior to the other two PID controllers in the face of torque disturbance especially in the yaw direction. The PID-position and PID-rate controllers initially have larger drifts when pitch and roll disturbances are present, although they can slowly stabilize the UAV. These phenomena may cause the UAV to move uncontrollably during takeoff, especially when the center of gravity is not near the center of the UAV. SMC has smaller drifts and corrects faster (see panels a, c, and t < 30 s). In the steady state, however, PID is more acceptable in terms of keeping the tracking error in x and y small. Note that, in the yaw direction, the other two controllers cannot quickly compensate for the disturbance (settling time out of range of the plot), so the PID-rate will slowly return to its original angle, while the PID-position will be slower. On the other hand, SMC is able to use the estimated disturbance information to form active compensation. In the case of circular wind or rotational forces acting on the UAV, SMC controllers with disturbance observers are more sensitive to torque than PID-based controllers, which results in SMC having stronger torque immunity.

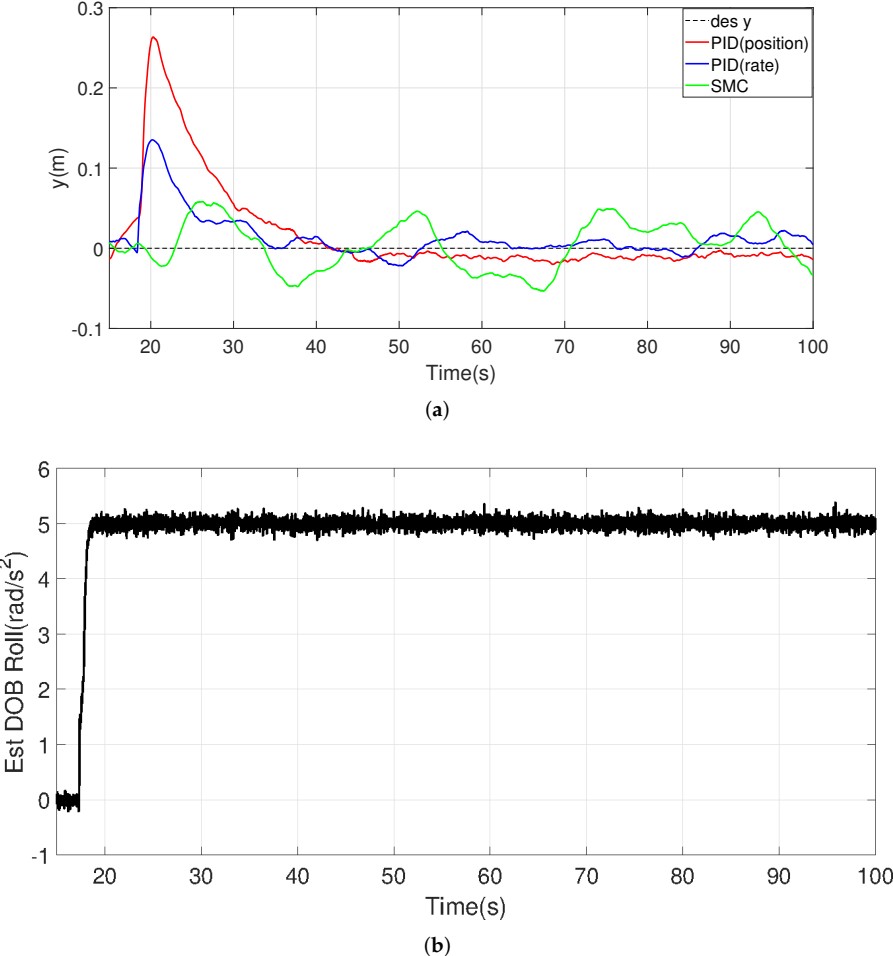

(a)

(b)

**Figure 22.** *Cont.*

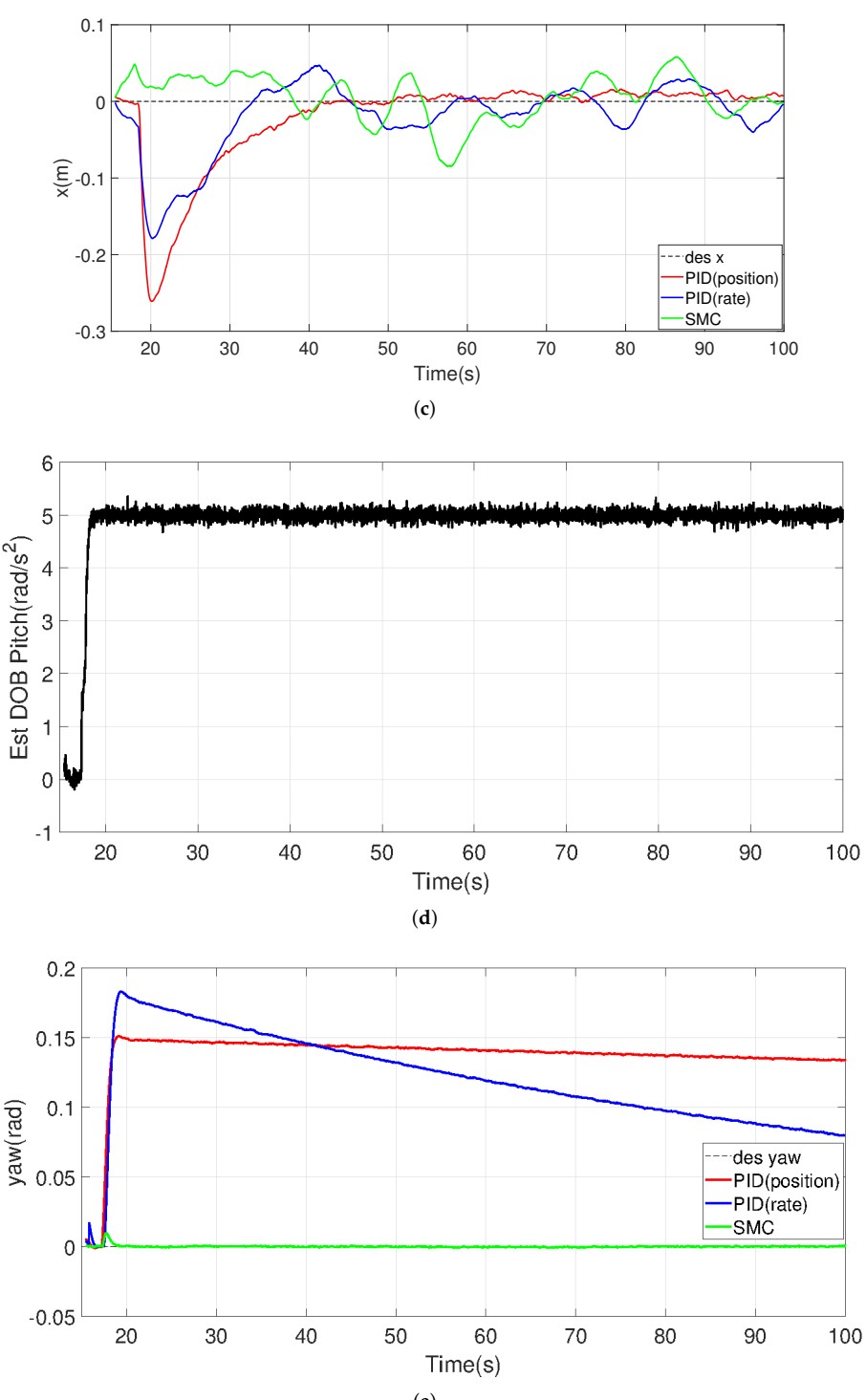

**Figure 22.** *Cont.*

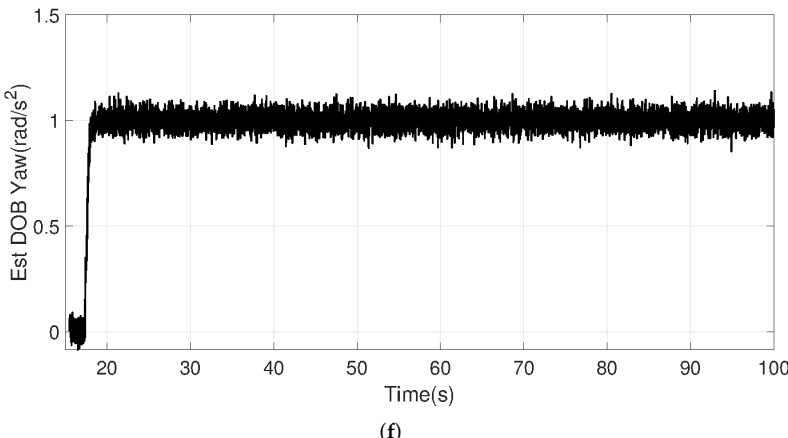

(**f**)

**Figure 22.** Origin-tracking results and disturbance estimation results under constant torque disturbance. (**a**) Y vs. Time. (**b**) Estimated disturbance in the $\phi$ Roll direction. Actual disturbance is 5 rad/s$^2$. (**c**) X vs. Time. (**d**) Estimated disturbance in the $\theta$ Pitch direction. Actual disturbance is 5 rad/s$^2$. (**e**) Yaw vs. Time. (**f**) Estimated disturbance in the $\psi$ Yaw direction. Actual disturbance is 1 rad/s$^2$.

## 4. Discussion and Conclusions

This work describes the design and implementation of a disturbance-observer-based sliding-mode surface controller (SMC) and validates a SMC with a simulated PX4 quadcopter. As a model-based design, this study started from a mathematical MATLAB/SIMULINK model, from which we adopted the PSO algorithm to optimize the parameters for the controllers, followed by testing and validating in a highly realistic jMavsim simulator. In the simulations, we compared two different benchmark PID controllers (PID-rate controller and PID-position controller) with the proposed SMC and evaluated their performance by applying different disturbances (e.g., disturbance-force, disturbance-torque, and crosswind effects.) within the sensor-noisy environment.

On the one hand, for precise motion in the horizontal plane, SMC is way ahead of the pack. On the other hand, the PID-based controller took more time to return to the reference value under sudden increases in force and torque disturbances, while the SMC could track the desired trajectory smoothly and quickly. In addition, when focusing on contour tracking under wind effects, both PID controllers performed similarly but were inferior to the SMC controller. This means that the SMC-based UAV has improved wind resistance. Another advantage of the SMC controller is that the sliding-mode controller was designed for a fully nonlinear model based on Lyapunov stability analysis; however, the PID controller was designed for a simplified linear form, which leads to a more robust SMC approach. An SMC with disturbance observers facilitates accurate and fast UAV adaptation in inconsistent dynamic environments.

## 5. Future Work

Future work can be divided into three parts. The first part is to optimize the proposed SMC controller to an actual UAV. No matter how superior the simulation results are, there may still be some discrepancies in reality. Therefore, it is necessary to repeat the tests on a real UAV to find the discrepancies and then improve and modify the controllers. The second part is to enrich the mathematical dynamic model. We will add more noise, simulate the process of signal transmission, and model more complex real-world environmental factors. The third part is to improve the SMC controller. In this study, we used a relatively basic form of a sliding surface; in the future, we will consider designing an adaptive sliding control law.

**Author Contributions:** Conceptualization, R.S. and J.M.-L.; methodology, Y.J., X.W., R.S. and J.M.-L.; software, Y.J. and X.W.; validation, Y.J., X.W. and C.F.; formal analysis, Y.J. and X.W.; investigation, Y.J., X.W. and C.F.; resources, R.S. and J.M.-L.; data curation, Y.J. and X.W.; writing—original draft preparation, Y.J. and X.W.; writing—review and editing, J.H.-J., R.S. and J.M.-L.; visualization, Y.J.; supervision, J.H.-J., C.G., R.S. and J.M.-L.; project administration, J.H.-J., C.G., R.S. and J.M.-L.; funding acquisition, J.H.-J., C.G., R.S. and J.M.-L. All authors have read and agreed to the published version of the manuscript.

**Funding:** This work is funded by the Air Force Research Laboratory.

**Data Availability Statement:** Not applicable.

**Acknowledgments:** The authors would also like to thank Weite Zhang, Chang, Liu, Xu Mao, and Yi Huang for their help in the simulations and experiments.

**Conflicts of Interest:** The authors declare no conflict of interest.

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
