# Peer review of "PX4 Simulation Results of a Quadcopter with a Disturbance-Observer-Based and PSO-Optimized Sliding Mode Surface Controller"

_drones, doi:10.3390/drones6090261_

Round 1

Reviewer 1 Report

In general revise English. In particular, avoid the use of “goog” “bad” “better” “worse “. In a paper language needs to be precise an accurate.

Reviewer 2 Report

The paper presents a nonlinear sliding mode surface controller (SMC) accompanied with a disturbance observer for a PX4 autopilot-based quadcopter. While some techniques regarding nonlinear sliding mode and disturbance observer have been applied in this study, the author wonders which is the novelty of this paper. A combination of some results from the previous studies seems not be sufficient. As the authors also mentioned in the inclusion, the proposed method should be evaluated with an actual drone and these experimental results should be added to enhance the results of the paper.
The affiliations of the authors are missed.

Reviewer 3 Report

Design Inner/Outer-Loop Sliding Mode Controllers with Disturbance Observer for PX4 Autopilot based Quadrotor

Authors:   Yutao Jing, Xianghe Wang, Juan Heredia-Juesas, Charles Fortner, Chris Giacomo, Rifat Sipahi, and Jose Martinez-Lorenzo

Summary:

In this paper, the authors proposed a new non-linear controller model called sliding mode surface controller (SMC) integrated with a disturbance observer for a PX4 autopilot-based quadcopter. They validated their model against the well established PID controller model used currently by the designers. Their study include developing a dynamic mathematical model and developing  an appropriate sliding mode control laws for each degree of freedom the quadcopter encounter. They optimized their SMC controller model parameters based on particle swarm optimization (PSO) method. Additionally, they experimentally evaluated and compared quadcopter’s tracking performance under a range of noise and disturbances including cross wind conditions. Their study concludes that the proposed SMC controller model facilitates accurate and fast UAV adaptation in inconsistent dynamic environments in comparison to the well-known PID controller models.  The authors are also included sufficient references related to the work presented. 

Recommendation: 

After reviewing the paper, it is recommended that this manuscript shall be acceptable for publication in MDPI-DRONES Journals with minor revision.  Some of the corrections identified are given below

Comments to Authors:

  1. The usage of grammar and English are good. However, there are spelling mistakes and sentence construction errors are noticed in few places. A general proofreading will improve the paper further.
  2. It is suggested to modify the title of the paper.
  3. It is suggested to rewrite the first sentence (This work…quadcopter) of the Abstract for clarity.
  4. It is suggested to  remove the word “extraordinary” in line 26.
  5. It is suggested to use small letter in “As” in line 87.
  6. The last paragraph of the introduction section needs to be rewritten since there are only 5 sections in the manuscript.
  7. In table 5, the unit for P is “us” or “μs”?
  8. The word “charity” needs to be corrected as “clarity” (line 171 and in line 412).
  9. It is suggested to replace the word “easily” with similar meaning word in line180, page 11, and two places in page 12.
  10. Some of the figures in the manuscript are in poor quality (or) it is difficult to read the text in the inside the figure and legends. Hence, it is recommended to replace the following figures with higher resolution figures. Fig. 5, Fig. 11, Fig. 14, Fig. 15, Fig. 16 (a) & (b), Fig. 19, Fig. 20, and Fig. 22.
  11. It is suggested to remove the word “dramatic” in line 257.
  12. It is recommended to rewrite the sentence “Note that…state” in line numbers 292-294 for clarity.
  13. It is recommended to rewrite the sentence “From the …deviation” in line numbers 349-361 for clarity.
  14. It is suggested to remove the word “some of you” in line 390 and it is recommended to rewrite this sentence for clarity.
  15. It is recommended to rewrite the sentence “This work…quadcopter” in line numbers 434-436 for clarity.
  16. In line 450, use the word “Another” instead of “another”.
  17. The section 5 details Future Work. Hence, it is recommended to change the heading for this section as Future work and move the Conclusions to previous section as “Discussion and conclusion”.
  18. As a general observation, it is noticed that some of the section and subsections headings are too short. Similar observations are also noticed for some of the captions in figures. Hence, it is recommended to update wherever necessary.

Round 2

Reviewer 2 Report

The reviewer thanks the authors for the responses. Most of the reviewer's comment has been clarified. However, since the present study evaluates the proposed method only by simulations, the authors should present some results regarding the stability of the control loop. Can the PSO algorithm ensure the stability of both systems using SMC and PID? In the experiment, is the PSO process implemented in real-time to update the parameter continuously? If it is that case, how about the computation cost of the PSO? 
